# Impacts of spatial resolution and representation of flow connectivity on large-scale simulation of floods

Cherry May R. Mateo[1,2], Dai Yamazaki[2,3], Hyungjun Kim[2], Adisorn Champathong[4], Jai Vaze[1], Taikan Oki[2, 5]

[1]CSIRO Land and Water, ACT, 2601 Australia
[2]Institute of Industrial Science, The University of Tokyo, Tokyo, 153-8505 Japan
[3]Department of Integrated Climate Change Projection Research, Japan Agency for Marine-Earth Science and Technology, Yokohama, 236-0001 Japan
[4]Royal Irrigation Department, Bangkok, 10300 Thailand
[5]United Nations University, 5 Chome-53-70 Jingumae, Shibuya, Tokyo 150-8925, Japan

*Correspondence to:* Cherry May R. Mateo (cherry.mateo@gmail.com)

**Abstract.** Global-scale River Models (GRMs) are core tools for providing consistent estimates of global flood hazard, especially in data-scarce regions. Due to former limitations in computational power and input datasets, most GRMs have been developed to use simplified representations of flow physics and run at coarse spatial resolutions. With increasing computational power and improved datasets, the application of GRMs to finer resolutions is becoming a reality. To support development in this direction, the suitability of GRMs for application to finer resolutions needs to be assessed. This study investigates the impacts of spatial resolution and flow connectivity representation on the predictive capability of a GRM, CaMa-Flood, in simulating the 2011 extreme flood in Thailand. Analyses show that when single downstream connectivity (*SDC*) is assumed, simulation results deteriorate with finer spatial resolution; Nash-Sutcliffe Efficiency coefficient decreased by more than 50% between simulation results at 10km resolution and 1km resolution. When multiple downstream connectivity (*MDC*) is represented, simulation results slightly improve with finer spatial resolution. The *SDC* simulations result in excessive backflows on very flat floodplains due to the restrictive flow directions in finer resolutions. *MDC* channels attenuated these effects by maintaining flow connectivity and flow capacity between floodplains in varying spatial resolutions. While a regional-scale flood was chosen as a test case, these findings should be universal and may have significant impacts on large- to global-scale simulations especially in regions where mega deltas exist. These results

demonstrate that a GRM can be used for higher resolution simulations of large-scale floods, provided that *MDC* in rivers and floodplains is adequately represented in the model structure.

## 1        Introduction

Catastrophes due to extensive, large-scale flood inundation have become more prevalent, especially in the last two decades (Brakenridge, 2015; EM-DAT, 2015). Although flood events typically occur locally, there is an increasing need for improved capability to predict flood inundation at large to global scales. Analyses at regional to global scales are essential to identify hotspots, provide consistent information for international financing of mitigation projects, and implement adaptation measures in a concerted and consistent manner (Döll et al., 2003; Adhikari et al., 2010; Pappenberger et al., 2012; Schumann et al., 2014). The simulations at regional to global scale are also necessary to determine specific locations for more in-depth analysis using detailed hydraulic and hydrodynamic models (e.g. MIKE FLOOD by DHI, 2005; TVD by Teng et al., 2015) which cannot be applied across very large areas.

Global River Models (GRMs, henceforth) are core tools for simulating flood hazards at regional to global scales. GRMs are necessary to simulate river discharge using runoff outputs from Global Circulation Models (GCMs), Global Hydrological Models (GHMs), or Land Surface Models (LSMs). Over the past decade, GRMs are increasingly being used to quantify global flood hazards and risks. Alfieri et al. (2013) used Lisflood Global (van der Knijff et al., 2010) to route the ensemble forecasts of surface and sub-surface runoff of HTESSEL (Balsamo et al., 2011) to produce global streamflow forecasts for early flood warning (GloFAS). Hirabayashi et al. (2008) integrated the TRIP GRM (Oki and Sud, 1998) with a GCM to project the global changes in flood and drought risks. The study was updated using a GRM with hydrodynamic representations (Yamazaki et al., 2011) and multiple climate models to project the future changes in flood risk under climate change  (Hirabayashi et al., 2013).

GRMs are typically structured to use the gridded outputs from GCMs, GHMs, or LSMs to simulate the lateral movement of water (Trigg et al., 2016). Formerly constrained by restrictive computational power and limited global datasets, GRMs are designed to use simplified representations of flow physics, use fewer, generalized parameters, and run at coarse spatial resolutions (Yamazaki et al., 2011; Neal et al., 2012b; Sood and Smakhtin, 2015). With the increasing computational power, improved computation algorithms, and availability of finer spatial datasets, it is now possible to model large- to global-scale floods in finer detail (Bierkens, 2015; Trigg et al., 2016). It is envisioned that the development and application of global models to finer spatial resolutions will lead to improvements in their accuracy and operational applicability (Wood et al., 2011; Lehner and Grill, 2013; Bierkens, 2015; Sampson et al., 2016). With respect to surface water and flood modelling, Wood et al. (2011) have highlighted the need for high spatial resolution to capture topographical controls which are critical for reliable and robust simulation of flood inundation.

Several studies have already demonstrated the possibility of modelling large- to global-scale flood hazards and risks at fine spatial resolutions using a cascade of models. Pappenberger et al. (2012) proposed and presented a proof-of-concept of a model

cascade composed of HTESSEL LSM (Balsamo et al., 2011) and CaMa-Flood GRM (Yamazaki et al., 2011) to derive consistent global flood hazard maps for 1 x 1km grids. Similarly, Winsemius et al. (2013) introduced GLOFRIS which derives flood hazards and risks at ~ 1km$^2$ using the PCR-GLOBWB GHM (van Beek et al., 2011) and DynRout, a global routing model similar to CaMa-Flood GRM but which uses a kinematic wave approximation (Petrescu et al., 2010). The modelling framework was implemented and validated in Bangladesh. A model cascade consisting of a regional rainfall-runoff model, a 1D diffusive wave river routing model, and 2D raster-based flood inundation model was used by Falter et al. (2016) to simulate flood risks at the Elbe River. Dottori et al. (2016) used the discharge outputs (available at 0.1° resolution) from the model cascade of GloFAS (Alfieri et al., 2013) to calculate the discharge maxima at several return periods. The discharge maxima were downscaled to 30" resolution and used as input to CA2D, a 2D hydraulic model (Dottori and Todini, 2011) to derive global flood hazard maps. Sampson et al. (2015) used a different approach by developing a regionalized flood frequency analysis that provides estimates of return period discharges from global datasets of stream gauging stations. The return period discharges were used as inputs to the sub-grid variant of LISFLOOD-FP (Sampson et al., 2013) to simulate high-resolution (~90m) global flood hazard maps.

While these recent studies successfully presented novel modelling approaches, there are several limitations common to them. First, while some of the studies employed sophisticated flood inundation models in their framework, most still require discharge outputs which are currently available at much coarser resolution (0.1 to 0.5 degree grids in a model comparison by Trigg et al., 2016) from a cascade of GHMs and 1D GRMs (e.g. works by Pappenberger et al., 2012, Ward et al., 2013, Winsemius et al., 2013, and Dottori et al., 2016) as inputs. Most of these studies identified the outputs from GHMs and GRMs as major sources of uncertainty. Falter et al. (2016) specifically pointed to the uncertainties coming from discharge simulations and 1D hydrodynamic simulations in river channels. Winsemius et al. (2013) pointed out that more focus should be given on studying the behavior of the GHMs and GRMs during extremes. Second, most of the results from these studies showed limited skill in simulating areas with multiple channel reaches such as floodplains and deltas. A study comparing several flood models found a significant difference in the simulated flood hazards in deltas in Africa (Trigg et al., 2016). GRMs typically assume that river channels flow to one downstream channel; this assumption can be an oversimplification of the more complex surface water flows in deltas and bifurcating channels (Yamazaki et al., 2014b), especially when applied to finer spatial resolutions. The limitations identified above can be potentially addressed with the use of an advanced GRM which can provide reliable predictions of flood discharge at finer spatial resolutions and can represent more complex surface water flows. The Catchment-based Macro-scale Floodplain model (CaMa-Flood) (Yamazaki et al., 2011, 2013, 2014b), is an advanced GRM which can be used in a cascade of model with GCMs, GHMs, or LSMs to calculate discharge, flood inundation, water level, velocity, and water storage. Estimation of flood characteristics on a regional to global scale is achieved by considering hydrodynamic flows and representing sub-grid topography in river channel and floodplains. Although the model uses a relatively simple 1D flow scheme which is commonly used in GRMs, CaMa-Flood is currently the only GRM that can simulate bifurcating flows in deltas and floodplains (Yamazaki et al., 2014b). While the model has been extensively validated and applied at regional and

global scales (see applications by Pappenberger et al., 2012, Hirabayashi et al., 2013, Mateo et al., 2014, Ikeuchi et al., 2015, and Trigg et al., 2016), its suitability for application to finer spatial resolutions is not yet verified.

The application of a GRM to finer spatial resolutions can potentially reduce the uncertainties that are incurred when downscaling coarser simulation outputs. More importantly, the application of a GRM to finer spatial resolutions will be beneficial in terms of capturing the effects of finer topographical controls on surface water flows. While global models are deemed to benefit from finer representation of topographical controls, predictability issues cannot be simply solved by finer resolution modelling – more focus should be given to address fundamental issues related to the realistic parameterization and appropriate representation of physical processes at the scale of application (Di Baldassarre and Uhlenbrook, 2011; Beven and Cloke, 2012).

In this study, the impacts of spatial scale and representation of flow connectivity between river channels and floodplains on the predictive capability of CaMa-Flood are investigated. With this verification exercise, we attempt to answer two fundamental questions with regards to large- to global-scale simulation of floods: (1) will the application of a GRM at fine spatial resolution provide better predictions, and (2) which flow processes should be represented in the model to realistically simulate flood discharge and inundation? CaMa-Flood is used to simulate a large-scale flood event which occurred at the Chao Phraya River Basin in Thailand at (1) varying spatial resolution, and (2) two flow connectivity representations. The test basin is introduced and the flood event described in the next section. A more detailed description of CaMa-Flood and the experimental setup is provided in the third section. Model calibration and validation is discussed in the fourth section. Quantitative and qualitative assessments of the results are presented in the fifth section. Lessons learned with regards to large-scale flood inundation modelling and the caveats of the study are discussed in the sixth section. The paper concludes with a summary of the findings and insights to the necessary development in GRMs towards finer resolution modelling of large- to global-scale floods.

## 2      Test Basin: Chao Phraya River Basin, Thailand

In this paper we assess the capability of CaMa-Flood to simulate a large-scale flood in the Chao Phraya River Basin in Thailand. The basin was chosen because of the complexity of its river network, and the recent occurrence of a large-scale flood.

With a catchment area of approximately 158,000 km$^2$, the Chao Phraya River Basin (shown in Fig. 1) is the largest and most important geographical unit in Thailand (Sripong et al., 2000). The northern region of the basin consists of mountainous areas; its middle region is a floodplain with a gentle slope of approximately 1/15,000, and its lowermost region is a delta. These geographic features make the region highly prone to flood inundation.

In 2011, enormous economic losses estimated at US$45 billion (World Bank, 2012) were incurred due to the extensive flooding of Thailand's industrial zones. The total rainfall during the rainy season was 143% (~ 1,439 mm) of that of the average rainy season from 1982-2002 (Komori et al., 2012); the flood peaks have exceeded the estimated 100-year return period peak discharges (DHI, 2012). The estimated 16-22 billion m$^3$ of flood volume inundated approximately 14 billion m$^2$ of its

floodplains (Rakwatin et al., 2013; Mateo et al., 2014). The extent of the damages affected the global supply chain of several industries, particularly the computer and automotive industries (Chongvilaivan, 2012; Swiss Re, 2012). The Thailand flood of 2011 is said to be the most economically damaging flood in recent history (Swiss Re, 2012; EM-DAT, 2015; Munich RE, 2013).

5    Two huge artificial reservoirs (Bhumibol and Sirikit) and several smaller artificial reservoirs are operational in the Chao Phraya River Basin. In this study, the impacts of reservoir operation on flood flows are removed by using naturalized flows (see Appendix for details) and assessing flood extents on dates when both reservoirs are already full and are assumed to have minimal impact on flooding.

## 3    Methods and Data

10  ### 3.1.    CaMa-Flood Model

CaMa-Flood model (Yamazaki et al., 2011, 2013, 2014b) was developed to realistically describe river routing considering floodplain inundation dynamics at the global scale. River basins are discretized and delineated at the desired spatial scale into unit-catchments, based on fine-resolution HydroSHEDS flow direction maps (Lehner et al., 2008) and SRTM3 Digital Elevation Models (DEMs) (Farr et al., 2007). Each unit-catchment is assumed to have a river and floodplain storage (Fig. 2a,
15  Table 1), the dimensions and characteristics of which are calculated using explicit sub-grid topography parameters derived from the fine-resolution flow direction maps and DEM. Gridded runoff from a GCM, GHM, or LSM is used as forcing input to calculate hydrodynamics at each unit-catchment within the river basin. River discharge and flow velocity along the river network at each unit-catchment are calculated using simplified shallow water equations. Water storage (water impounded in river and floodplains) at each unit-catchment, the only prognostic variable, is calculated and updated so that water mass is
20  conserved. From the water storage at each time step, other flood inundation characteristics such as water level and inundated area are diagnosed.

The latest and most efficient version of CaMa-Flood (Yamazaki et al., 2014b) which uses the "local inertial equation" (Bates et al., 2010) to calculate discharge at each unit-catchment was utilized in this study. By neglecting only the advection term of the 1-D St. Venant equation, the local inertial equation explicitly represents backwater effects and improves the representation
25  of shallow water physics in the model. According to Bates et al. (2010), the local inertial equation can be discretized and modified to Eq. (1),

$$Q^{t+\Delta t} = \frac{Q^t - \Delta t g A S}{\left(1 + \frac{\Delta t g n^2 |Q^t|}{A h^{4/3}}\right)} \tag{1}$$

where $Q^{t+\Delta t}$ is the discharge between times $t$ and $t + \Delta t$, $Q^t$ is the discharge at the previous time step, $g$ is the gravitational acceleration (m s$^{-2}$), $A$ is the flow cross-sectional area (m$^2$), $S$ is the water surface slope between the upstream and downstream
30  unit catchments, $n$ is the Manning's friction coefficient (m$^{-1/3}$ s), and $h$ is the flow depth (m).

A new flow scheme which uses an algorithm to identify and represent diverging channels in a fine-resolution river network map is incorporated into the latest version of CaMa-Food (Yamazaki et al., 2014b). By representing the more complex, diverging flows in deltas and floodplains, the new scheme overrules the simplified single downstream connectivity (*SDC*) assumption adopted in most 1D GRMs. Originally intended to simulate the bifurcation processes in deltas, the new scheme is referred to as the "bifurcation scheme" and the pathways which allow multiple downstream flow as "bifurcation channels" in the paper by Yamazaki et al. (2014b). Bifurcation channels, defined as channels connecting two unit-catchments which do not have upstream-downstream relationships in a river network map, are classified as either "overland pathways" (green lines in Fig. 2b) or "river pathways" (red lines in Fig. 2b).

The algorithm for extracting bifurcation channels is described in detail in Yamazaki et al. (2014b) and will only be described briefly in this paper. Using data from HydroSHEDS and SRTM3, the algorithm searches for possible flow pathways which cross unit-catchment boundaries. A "bifurcation threshold height" above the main channel of each unit-catchment is set for computational efficiency. The algorithm searches for pixels (grid cells in the SRTM3 DEM) which are at unit-catchment boundaries and are at an elevation lower than that of the bifurcation threshold. The pixel is identified as a valid bifurcation point if its elevation is higher than that of an adjacent pixel which is located in another unit-catchment. Using HydroSHEDS flow directions, a bifurcation channel is defined as the pathway from each bifurcation point to the main channel pixels of its upstream and downstream unit-catchments. Bifurcation channels in floodplains are represented by overland pathways, while those with persistent bifurcated flow are represented by river pathways. Persistent bifurcated flow is detected using the SRTM Water Body Data (SWBD) water mask (NASA/NGA, 2003). By representing bifurcation channels as described above, flows in braided streams, artificial open canals, diversion channels, and other diverting water pathways can be represented in flood simulations. Hence, the new scheme not only allows the simulation of flows in bifurcating rivers – generally, it enables the simulation of Multiple Downstream Connectivity (*MDC*) between grid cells. To avoid misconceptions about the function of the new scheme, it will be referred to as "*MDC* scheme" hereafter. For simplicity, channels or flow pathways which enable *MDC* are referred to as "*MDC* channels" or "*MDC* pathways."

Flow in a *MDC* channel occurs when the elevation of the channel is lower than the water surface elevation in either upstream or downstream unit-catchment. Flows in *MDC* channels are calculated using the local inertial equation after the flows in main channels have been calculated (Yamazaki et al., 2014b).

The development of CaMa-Flood model is well-documented and its performance, well-validated. For more details about the model, please refer to the papers describing its development (Yamazaki et al., 2011, 2013, 2014b).

### 3.2.    Experiment Setup and Input Data

The FLOW algorithm (Yamazaki et al., 2009) was used to upscale the river network map and calculate the sub-grid river channel and floodplain topography in the Chao Phraya River Basin at the following spatial resolutions: 30-arcsecond (~1km), 1-arcminute (~2km), 2-arcminute (~4km), 3-arcminute (~6km), 4-arcminute (~8km), and 5-arcminute (~10km). The river and sub-grid parameter maps were extracted from the 3-arcsecond HydroSHEDS flow direction map (Lehner et al., 2008; Lehner

and Grill, 2013) and SRTM3 DEM (Farr et al., 2007). Simulations were performed at each spatial resolution switching channel connectivity representation between the *SDC* and *MDC* scheme in CaMa-Flood.

To focus on the impacts of spatial resolution and flow processes on the river model, the same daily runoff dataset with a spatial resolution of 5-arcminute square grids was used as forcing input to CaMa-Flood. To conserve the mass of runoff inputs, CaMa-Flood uses area-weighted averaging to distribute the coarse, gridded runoff among the unit-catchments in CaMa-Flood. The gridded daily runoff dataset was simulated using the land surface processes module of H08 integrated water resources model (Hanasaki et al., 2008a, 2008b; Mateo et al., 2014). The runoff data was simulated using the following meteorological forcing: surface air pressure, wind speed, specific humidity, shortwave radiation, longwave radiation, temperature, and surface albedo from a study by Yoshimura et al. (2008), and precipitation data reanalyzed from gauge stations provided by the Royal Irrigation Department (RID) and Thai Meteorological Department (TMD) of Thailand. For further information regarding runoff simulation, please refer to Mateo et al. (2014).

The simulation domain was set from 97°E to 102°E longitude and 13°N to 20°N latitude. The calculation time step was automatically adjusted by the Courant-Friedrichs-Lewy condition (see Bates et al., 2010; Yamazaki et al., 2013) in CaMa-Flood. The simulation period was set from 2010 to 2011, with one year spin-up period.

## 4    Model Calibration and Validation

The sub-grid river cross-section and channel roughness parameters calibrated by Mateo et al. (2014) were used in this study. Parameter tuning and validation were executed by comparing simulation outputs with the surveyed river cross-section and observed river discharge data provided by the RID. Observed satellite images obtained by combining the Moderate Resolution Imaging Spectroradiometer (MODIS) and Advanced Microwave Scanning Radiometer for EOS (AMSR-E) products were used to further tune the sub-grid channel parameters and validate the simulated extent of flood inundation (obtained through personal communications from Dr. Wataru Takeuchi of the University of Tokyo). These images are available in 10-day intervals at a spatial resolution of 10-arcmin (~20km).

The parameterized sub-grid river and floodplain topography as calibrated by Mateo et al. (2014) are shown in Eq. (2) and (3),

$$W = \max\left[16.6 * R_{up}^{0.35}, 3.0\right] \tag{2}$$

$$B = \max\left[0.70 * R_{up}^{0.23}, 0.20\right] \tag{3}$$

where W is the river width, B is the river bank height, and $R_{up}$ is the annual maximum 30-day moving average runoff from upstream of the unit-catchment (Yamazaki et al., 2011). Based on Mateo et al. (2014), the Manning's coefficient was fixed at 0.024 in the river channel and 0.10 in the floodplain for the entire basin. These values of Manning's coefficient are comparable with those used in other studies in the Chao Phraya River Basin (Visutimeteegorn et al., 2007; Keokhumcheng et al., 2012; Sayama et al., 2015) and those obtained by USGS from lab and field data (Aldridge and Garrett, 1973). By using the calibrated parameters, CaMa-Flood can adequately simulate the discharge and flood inundation in the basin (Fig. 3) (Mateo et al., 2014).

The model calibrated at 10 km spatial resolution was found to have a good fit with observations. The discharge estimates from the model were in good agreement with the observed river discharge at the station used for calibration (C2 Station in Fig. 1), with the daily Nash-Sutcliffe efficiency coefficient (NSE) in year 2011 with SDC and MDC of 0.73 and 0.80, respectively. The Pearson correlation coefficients between the observed and model estimated discharge are very high (both above 0.90) and biases are low. There is also a very good agreement between the model estimated flood inundation extents and the satellite derived water maps for all available satellite images. Validation of the model for different years and other gauging stations in the Chao Phraya River Basin are also shown to be reasonable by Mateo et al. (2014). The results of the calibration confirm that the parameterisation is reasonably robust and suitable for large scale application in the Chao Phraya River Basin. This is to be expected as even without calibration of the parameters, the use of CaMa-Flood with nine GHMs (which include the H08 model) results in better agreement with monthly to daily observations in 1701 globally distributed river discharge stations from the Global Runoff Data Center as compared with the native river routing schemes of the GHMs (Zhao et al., 2017).

The calibrated parameters were perturbed and applied at a finer spatial resolution to examine if recalibration or tuning of the parameters is needed to apply the model at finer spatial resolutions. It was found that the 'optimum' parameters do not significantly change with changes in spatial resolution (see Appendix). Hence, the parameters of the model which were calibrated at coarser scales can be applied at finer scales without recalibration to avoid huge computation overheads. The stability of the calibration across scales indicates that the model is robust.

## 5    Results

This section starts with a discussion of the impacts of spatial scale and representation of *MDC* on the predictive capability of the model. Subsequently, to explain the causes of the changes in model efficiency, the impacts of scale and representation of *MDC* on channel topography and flood dynamics are discussed. For brevity, hereafter, simulations with higher spatial resolution are referred to as "finer resolutions" or "increased resolutions," and simulations at approximately *n* km spatial resolution as *n*k (e.g., 1k for 30-arcsecond or ~1km at the equator).

For better visibility when analyzing spatial impacts, we zoomed into the three areas (indicated by the boxes in Fig. 1) located in the upper (up), middle (mid), and lower (low) sections of the catchment. For brevity, only the results in the lower sections of the catchment are shown in this paper.

While the results shown in the following subsections are obtained by running the model with calibrated parameters, numerical experiments were also conducted by running the model using alternative parameter values (e.g. parameter values used in global simulations, using the river widths from the global width database for large rivers, GWDLR by Yamazaki et al. 2014a). The findings remain the same (and thus are not shown in this paper for brevity) albeit there are differences in the magnitude of changes in model efficiency with changes in spatial resolution. As expected, the use of non-calibrated parameters resulted in larger changes in model efficiency. Hence, the findings of this study are robust and are independent of the parameters used in the model.

### 5.1. Impacts on predictive capability of the model

Six metrics were used to objectively evaluate the predictive capability of the model in each simulation setup: NSE, root mean square error (RMSE), Pearson correlation coefficient (correlation), percent bias (PBIAS), difference in discharge peak timing (in days), and spatial measure of fit of the flood extent (F in Eq. (4), Bates and de Roo, 2000),

$$F = 100 * \frac{Num(S_{mod} \cap S_{obs})}{Num(S_{mod} \cup S_{obs})} \qquad (4)$$

where $S_{mod}$ and $S_{obs}$ are unit-catchments (or pixels) which are flooded in the model and satellite observation, respectively. The first five metrics evaluate the capability of the model to simulate discharge at the 11 gauge stations (white dots in Fig. 1) while the last evaluates the capability of the model to predict the spatial extent of inundation.

To reduce the impacts of human intervention on the calculated model efficiencies, the observed discharges were naturalized as necessary (see Appendix) and flood extent was compared on dates when the dams were filled to their capacity (i.e. minimal effect on downstream flows). Daily discharges in the entire year of 2011 were used to calculate the flow metrics. The flood extent for the month of October 2011 were used to calculate the fit of flood extent, F. To ensure comparability, the simulated flood volumes were projected on the fine-resolution DEM and aggregated to the resolution of the observed satellite images.

It was found that the statistics related to model efficiency do not significantly change in most of the upstream gauging stations. This can be due to at least one of following reasons: 1) the unit-catchment is located near or within a mountainous area where bank slope is relatively high and where kinematic wave processes govern more than other flood dynamic processes, and/or 2) *MDC* pathways are not prevalent and do not significantly affect the simulated flows or inundation. Figure 4 shows that majority of the validation stations are located in regions that have relatively low density of *MDC* pathways. Hence, the discussions in this section will focus where the impacts of spatial resolution and complex flows are significant – stations C3 and C13 located in the low-lying floodplain areas.

The changes in model efficiency are more evident in *SDC* simulations: between simulation in 10k and 1k resolutions (Fig. 5a to 5e), NSE and correlation of discharge drastically declined by as much as 50% and 10%, respectively, while RMSE, PBIAS and difference in discharge peak timing drastically increased by as much as 90%, 35%, and 70%, respectively. On the other hand, model efficiency incrementally increased and errors marginally decreased with finer resolutions in *MDC* simulations (below 3% change between simulations in 10k and 1k). Although statistically insignificant (less than 5% between simulations in 10k and 1k), both *MDC* and *SDC* simulations show increasing trends in the fit of flood extent (F shown in Fig. 5f) with finer resolution. In all the six metrics, *MDC* simulations consistently outperformed *SDC* simulations.

### 5.2. Impacts on river network and channel topography

The river maps (elevation, width, upstream area, and other river channel and floodplain properties) used for simulation are identical between flow connectivity schemes. *MDC* pathways were only added to the river maps used for *MDC* simulations.

As have been shown in Fig. 4, the number of detected and represented multi-directional pathways increases as resolution increases.

The impacts of varying spatial scale to the river network and river channel characteristics is summarized in Table 2 and shown in Fig. 6. As expected, topography and river network in the river basin are better represented in the finer resolution maps (see Fig. 6a). Figure 6b shows evident changes in the distribution of small streams represented in the model while those of main channels do not significantly change.

The equations used to parameterize the sub-grid river widths and bank heights (see Eq. (2) and (3) in the previous section) are dependent on a fixed minimum width/bank heights and the annual maximum runoff in the sub-catchment area upstream of the unit-catchment (Yamazaki et al., 2011). Because the same runoff data (GRDC-based for the parameterization) was used to generate the river maps, the sub-grid topography of river channels which have been identified in coarser resolution maps did not significantly change in finer resolution maps. These river channels typically have large cross-sectional areas (river width 100.0m, river bank height 2.0m; called main channels hereafter). However, the discretization process resulted in an increase in the proportion of small river channels (river width < 100.0m, river bank height < 2.0m; called main small streams hereafter) represented in finer resolution maps.

## 5.3.     Impacts on flood inundation and flow characteristics

Generally, the increasing discretization of the river basin in finer resolution maps led to increasingly defined flood depths and flood extents (see Fig. 7) and hence, the increasing fit of flood extent, F, in both *SDC* and *MDC* simulations. This can be mainly attributed to the more defined representation of topographical details in finer spatial resolutions.

Flooded unit-catchments in *SDC* simulations tend to be confined near main channels. The "one-downstream-grid" assumption used in *SDC* simulations constrained the flow of water within sub-catchments that have upstream-downstream relationships. This resulted in unrealistic flood inundation patterns and water surface elevation in *SDC* simulations: water surface elevation in a very flat portion of the river basin shown in Fig. 7e and 7f reveal unlikely flood boundaries and abrupt drops of more than 5 meters in elevation.

These effects are avoided by allowing water flows between adjacent floodplains that do not have explicit upstream-downstream channel relationships through *MDC* pathways. As a result, more widespread flood extent with more realistic, gradually decreasing water surface elevation is obtained in *MDC* simulations (Fig. 7g and 7h). Flood inundation in unit-catchments with main channels became less severe in *MDC* simulations than those in *SDC* simulations (last two columns and first two columns in Fig. 7, respectively).

The simulated velocity shows an increasing number of unit-catchments with backflow (negative velocity shown in pink pixels in Fig. 8a to 8d) with finer resolution. The negative flows in *MDC* simulations have lower magnitudes and were more intermittent as compared with those in *SDC* simulations. In effect, flood duration became longer in finer resolutions in *SDC* simulations, as shown in Fig. 8e and 8f. The simulated hydrograph at C13 station shown in Fig. 8i reveal decreasing outflows with increasing resolution in the rising limb and a reversal of this trend in the recession limb. The difference in time to peak

with increasing resolution is also evident in the *SDC* simulated hydrographs. Such patterns are subdued or not evident in *MDC* simulations (Fig. 8j).

**6       Discussion**

**6.1.      Importance of representing multiple downstream connectivity**

Most global flood models use discharge outputs from a coupling of GHMs or GCMs and 1D GRMs in their cascade of models (e.g. ECMWF by Pappenberger et al., 2012, GloFRIS by Ward et al., 2013 and Winsemius et al., 2013, JRC by Dottori et al., 2016). Before the development of CaMa-Flood with *MDC* scheme, runoff in most GRMs is routed throughout the land mass by discretizing the river basin into grids or smaller sub-catchments, where each grid or sub-catchment is assumed to have one river channel that flows to one downstream channel (*SDC* assumption). This means that an upstream-downstream relationship between two grids is necessary for water to flow between them. By using this simplified approach, runoff generated by GHMs or GCMs can be routed throughout the basin and can be calculated in large domains.

The older generation of GRMs which use the *SDC* assumption were designed to simplify the representation of network and flows in continental rivers and to run in relatively coarse spatial resolutions (coarser than 10km grid resolution). In coarse spatial resolutions, one grid cell may be large enough to cover an area with a river delta (its main channel and braided streams or tributaries). In such cases, the grid cell may be assumed to flow towards one direction, most likely towards the direction of the next main channel; hence, the *SDC* flow scheme may be sufficient to represent the river network and flows realistically. However, this assumption may be too simplified or inappropriate for representing the flow network in deltas and braided streams when we move to finer spatial resolution simulations.

As had been shown in section 5.2, finer resolution modelling results in higher discretization of the river basin which then results in more rivers, particularly small streams, to be represented (see schematic diagram in Fig. 9a and 9b). While the finer discretization and better representation of small rivers may result in more realistic representation of the river network in hilly or mountainous areas, it will also result in the disaggregation and reduced connectivity in floodplains. A grid with a wide floodplain connected to a main channel in coarse resolution will be discretized into smaller grids with "small river" channels that flow towards the main channel. A coarse resolution grid which contains a main channel that flows to diverging smaller streams will be discretized into smaller grids with the main channel grids flowing towards a downstream main channel grid, and the smaller streams disconnected from the main channel (no inputs from the main channel) and flowing towards a downstream small stream grid.

In the event of flooding, floodplains in coarse resolution simulations (left-most column, Fig. 9a to 9c) will be filled with water that is at the same level as that of in main channels. Water in floodplains flow in the same direction as the main channel (yellow orange arrows in Fig. 9). On the other hand, in finer resolution *SDC* simulations, the floodplains are discretized into unit-catchments with "small streams" (teal cross-section of the middle figure in Fig. 9d). These "small streams" will be filled with water which can only flow through its small channel towards (or away from) the main channel; water cannot flow in the same

direction as the main channel (brown arrows in Fig. 9d). In effect, the capacity of floodplains to allow water to flow through (hereby referred to as flow capacity) to the downstream channel is significantly reduced in finer resolution *SDC* simulations. Multi-directional flow connectivity reduces the effect of this phenomenon. With the presence of *MDC* pathways, water in "small streams" can flow both towards the main channel and the direction of the *MDC* pathway (purple arrows in Fig. 9);

hence, similar flow capacity as in coarser simulation can be maintained in finer *MDC* simulations.

In the occurrence of a flood wave, water level in the main channels in finer resolutions will become higher than the upstream "small rivers." When this occurs in *SDC* simulations, water can only flow backwards in the small rivers. This explains the severe and widespread "build up" of backflows in *SDC* simulations in Fig. 8b, and the "trapping" of water which lead to longer simulated flood days in Fig. 8f. The massive backflows are avoided in *MDC* simulations (evident in Fig. 8c, 8d, 8g, and 8h)

because *MDC* pathways allow water to flow to surrounding grids other than the main channel (purple arrows in Fig. 9).

These results stress the importance of representing multiple downstream flow connectivity in finer resolution simulation of large-scale floods, particularly in floodplains. While sub-grid parameterization of river and floodplain topography should be enough to realistically constrain the flood extent and simulate the flood depth in areas with high to moderate slopes, single-downstream-grid river channel and floodplain flows will not be able to realistically simulate the spreading of water in very flat

floodplains and deltas. *MDC* pathways are not only useful for representing braided or bifurcating streams in coarser resolution simulations – they are necessary for representing and maintaining flow connectivity and flow capacity in very flat floodplains in finer resolution simulations.

### 6.2. On improved simulation of large-scale floods in finer resolutions

Small catchment-scale topographic controls and flow processes are critical in regulating the storage and movement of surface

waters (Yamazaki et al., 2011; Neal et al., 2012b). While finer resolution modelling leads to improved representation of topography, simulation of large-scale floods using GRMs will not necessarily improve through hyper-resolution modelling alone – GRMs will have to be improved by sufficient representation of flow physics (both sub-grid and between grids) which are appropriate for the scale of implementation.

In the test basin used, the realistic representation of flow connectivity and flow capacity in floodplains was found to be critical

in finer resolution modelling of large-scale floods. Although in reality all flows occur in 3D, the relatively reduced complex modelling structure implemented in CaMa-Flood was found to be sufficient in simulating extensive flooding in the test basin. CaMa-Flood sufficiently represents both topographic controls and flow processes in large-scale simulations through: (1) sub-grid parameterization of river and floodplain topography, (2) use of local inertial equation in calculating 1D channel and floodplain flows, and (3) representation of multiple downstream connectivity of flows.

While the application of GRMs in a cascade of large- to global-scale models to finer spatial resolution is ideal, it is important to note that they are not meant to replace catchment-scale flood models. Where more detailed local data is available, catchment-scale hydrodynamic models are more suitable for thorough planning and exploration of management and mitigation options at local scales (Ward et al., 2015; Teng et al., 2017). Catchment-scale hydrodynamic models usually have a more complete

representation of surface water flow physics as compared to global models. Although more detailed hydrodynamic models can simulate flood processes more accurately, their implementation in regional to global domains entails high computational costs; the models either have to be implemented in coarser resolutions or re-configured to simplify their representation of flow processes (Horritt and Bates, 2001; Hunter et al., 2007). The predictive capability of hydrodynamic models deteriorates when implemented in coarser resolutions (Kirchner, 2006; Neal et al., 2012a, 2012b). The governing physical equations in these models are not applicable to more complex, heterogenous systems in larger domains (Kirchner, 2006) and most of the key topographic controls within the floodplain are smeared when aggregated to coarser scales. While several catchment-scale hydrodynamic models have already been applied to large river basins by representing sub-grid topographical controls in coarser resolutions (e.g. Wilson et al., 2007; Neal et al., 2012b), such application still requires a significant amount of boundary data processing and computational resources when coupled with GCMs and GHMs (Yamazaki et al., 2014b). Such applications of catchment-scale hydrodynamic models to coarse spatial resolution simulation of large-scale floods are not shown to be superior to those from advanced GRMs. To harness the benefits from using GRMs and catchment-scale hydrodynamic models, the development of hybrid approaches, where outputs from CaMa-Flood with *MDC* scheme are used as initial or boundary conditions of catchment-scale hydrodynamic models, may be developed and assessed in the future. Hybrid approaches using relatively simpler GRMs have been shown to be feasible in the continental to global scale mapping of flood hazards and risks in fine spatial resolution (e.g. Ward et al., 2013, Winsemius et al., 2013, and Dottori et al., 2016).

### 6.3.    Caveats and future works

One of the caveats of this study is the tedious calibration of the sub-grid channel parameters when applied to regional basins. At the global scale, this can potentially be addressed by the global width database available for large rivers (GWDLR developed by Yamazaki et al., 2014a). A database of channel depths of large rivers, however, do not exist; hence, the parameters characterizing the channel depths in the model may still have to be calibrated. To ease the difficulty of calibration, the development of an automatic calibration tool or a simpler or more efficient parameterization of sub-grid channels (e.g. Neal et al., 2015) may be helpful.

In the test basin used in this study, it was found that the parameters calibrated at a coarse spatial resolution are transferable across finer spatial resolutions. This significantly reduces the time required to re-calibrate the model at finer spatial resolutions. Once the initial difficulty of calibrating the necessary parameters at a coarse spatial resolution is hurdled, CaMa-Flood with *MDC* scheme can be used for more realistic, consistent, and robust simulation of large-scale floods across varying spatial resolutions. It should be noted, however, that the *MDC* scheme of CaMa-Flood had only been validated in three test basins – Mekong delta (Yamazaki et al., 2014b), Ganges-Brahmaputra-Meghna delta (Ikeuchi et al., 2015) and Chao Phraya River Basin in Thailand (this study). It should also be noted that the transferability of calibrated parameters from a coarse spatial resolution to finer spatial resolutions have to be validated in other river basins. More extensive tests in large river basins and at global scale have to be conducted to further validate the model.

While the use of the *MDC* scheme in CaMa-Flood resulted in improvements in the simulation of flood dynamics in large floodplains, it should be noted that uncertainties remain in the representation of *MDC* pathways in the model. The *MDC* pathways in the model may not necessarily correspond to or explicitly represent the actual flow pathways, especially the small channels (e.g. small artificial canals) in the river basin. Small *MDC* channels in the model which are not covered by the SWBD water mask are currently represented as overland pathways. As channel bathymetry is not considered in overland pathways, this assumption may lead to the underestimation of flows in small *MDC* channels in the model (Yamazaki et al., 2014b). The accurate representation of *MDC* pathways in the model still depends on the resolution and accuracy of the DEM used (3-arcsecond or ~90m SRTM3 DEM by Farr et al., 2007 in this study). The explicit representation of small artificial channels with widths which are narrower than the grid resolution of the DEM used and other small scale flow connectivity between rivers and floodplains is still difficult to achieve in large scale simulations. A finer resolution DEM or a tool for extracting or deriving smaller channels from open street maps will be very helpful in improving the accuracy of the representation of *MDC* pathways in the model. The use of new data-driven approaches to derive flood-mediated *MDC* pathways and connectivity (e.g. progressive nearest neighbour search or progressive iterative nearest neighbour search by Zhao et al., 2017) may also be explored in the future. It should also be noted that the changes in channel network (by sedimentation, subsidence, and other geological processes, or by levee breaks, water diversion, and other anthropogenic impacts) and operation of artificial canals are not represented in the current model. Such natural or anthropogenic influences which add to the complexities in real flow pathways have significant impacts on the connectivity and flood dynamics in floodplains (Syvitski et al., 2005; Alsdorf et al., 2007; Schumann et al., 2011; Trigg et al., 2013). However, even catchment scale hydrodynamic models implemented in fine spatial resolution have difficulties in representing such complex processes. The representation of such complexities will require the integration of more detailed models and data (e.g. landscape, sedimentation, anthropogenic, etc.) with flood models.

Other than representing *MDC* in a 1D model, there are other model structures which can realistically simulate flood dynamics in large floodplains such as those that implement full St. Venant momentum equations (e.g. Paiva et al., 2011, 2013). For the benefit of developing reduced complexity models which can adequately simulate large-scale floods in finer resolution, more model structures should also be assessed in the future.

The limited availability of inundation data for validation of flood extent at large-scales, especially in data-poor regions, is another caveat of this study. The satellite-derived inundation product used in this study is too coarse to comprehensively assess the capability of the model to simulate flood inundation parameters at varying spatial resolutions. Better flood satellite images available at large-scales and in finer spatial and temporal resolutions will certainly be beneficial for such analyses.

The relatively coarse runoff forcing data is also another issue that needs to be addressed. Similar with most studies done in the past (e.g. Kumar et al., 2006; Famiglietti et al., 2009), this study is largely constrained by the lack of meteorological forcing data sets at finer resolution. The authors tested the use of finer resolution runoff data generated from simple spatial interpolations with topographical correction in the meteorological data, but no significant results have been found (and hence, were not shown in this study for brevity). However, finer resolution runoff data generated from better and finer inputs can potentially improve results in other river basins. Previous studies have already shown that using finer spatial resolution inputs

could result in improvements in runoff and water balance calculations (Kumar et al., 2006; Singh et al., 2015). In this regard, a study which involves varying resolution of runoff inputs, sub-grid river and floodplain topographical maps, and model implementation may be conducted in the future.

### 7    Conclusion

In this study, we have assessed the suitability and demonstrated the capability of an advanced GRM for simulating flood discharge and inundation at finer spatial resolutions. Both the impacts of more realistic representation of downstream flow connectivity and finer spatial resolution have been assessed. While the predictive capability of the model improved slightly with finer spatial resolution when multiple downstream flow is considered, it declined significantly when single downstream flow, the flow connectivity scheme used in most GRMs, isused. To keep the level of simulation skill of a GRM at finer spatial resolutions, appropriate set of physical representations should be included in the model. In this study, it was found that representing multiple downstream flow connectivity is important in the realistic simulation of inundation in floodplains, especially in finer spatial resolutions. *MDC* pathways provide two essential functions in simulation of large-scale flood inundation: (1) represent flow connectivity between floodplains which allow more realistic flood routing in deltas and low-lying flat lands, and (2) maintain the flow capacity in floodplains and river channels across varying spatial scales. These results clearly show that with regards to large-scale modelling of flood inundation in very flat floodplains and deltas, (1) finer resolution modelling will not always result in better predictions, and (2) multi-directional flow connectivity is one of the important flow processes that have to be represented in the model structure.

The findings of this study demonstrate the limitations of GRMs which cannot realistically represent floodplain connectivity and follow the "one-downstream-grid" assumption. These results are indicative of the flow processes necessary in GRMs and global flood models to adequately predict flood discharge and inundation at finer spatial resolution. This study also provides insights for GRM developers targeting application at multiple scales.

**Appendix: Methodology for and stability of calibration**

There are several long-standing issues regarding parameter calibration in flood inundation models. Parameters that give a high efficiency/correlation NSE do not necessarily result in low errors in RMSE/PBIAS nor high predictive capability of simulating flood extent; multiple good parameter sets may also exist (equifinality problem) (Beven, 2002; Pappenberger et al., 2005; Yamazaki et al., 2011). Hence, the calibrated parameter set may give good overall results but may not always perform best in all measures of efficiency. Therefore, the focus of this section is not to test whether the calibrated coefficients will result to optimal performance in all metrics; rather, it examines whether the optimal coefficients in one spatial scale remain to be so in the other spatial scale.

CaMa-Flood and H08 models were calibrated by increasing the NSE-coefficient in monthly and daily discharges, reducing the errors in the timing and magnitude of peak discharge during the worst drought and worst flood year in the dataset, and comparing the spatiotemporal characteristics of flood extent. The calibration of parameters using discharge was carried out at Nakhon Sawan Station, also known as C2 Station (marked by an orange dot in Fig. 1), a gauging station critically located just after the confluence of the four main tributaries of the Chao Phraya River Basin. The simulated daily and monthly discharge hydrographs are compared with that of the naturalized observed to remove the effects of the operation of Bhumibol and Sirikit reservoirs. The naturalized observed discharge was computed by deducting the effects of reservoir operation upstream of the station using Eq. (A1),

$$ND_{C2} = OD_{C2} + [I + P - R - S]_{Bhumibol} + [I + P - R - S]_{Sirikit} \qquad (A1)$$

where ND is naturalized discharge, OD is observed discharge, I is reservoir inflow, P is water pumped into the reservoir, R is reservoir release, and S is water released through the spillway. Gauging stations within the basin with observation years greater than 10 and catchment area greater than 10,000 km$^2$ have been chosen for the validation (marked by white dots in Fig. 1). Similarly, the observed discharge in validation stations downstream of either Bhumibol or Sirikit Reservoirs were naturalized using equation A1 modified accordingly before comparing with the simulated discharge. For further details on calibration and validation, please refer to Mateo et al. (2014).

CaMa-Flood model can be calibrated and tuned by changing coefficients $x_w$, $y_w$, $z_w$, $x_b$, $y_b$, and $z_b$ in equations A2 and A3:

$$W = \max\left[x_w * R_{up}^{y_w}, z_w\right] \qquad (A2)$$

$$B = \max\left[x_b * R_{up}^{y_b}, z_b\right] \qquad (A3)$$

As previously mentioned, W is the river width, B is the river bank height, and $R_{up}$ is the annual maximum 30-day moving average runoff from upstream of the unit-catchment (Yamazaki et al., 2011). The model is quite sensitive to the sub-grid channel parameters. A sensitivity analysis done by Yamazaki et al. (2011) showed that a deeper bank height, wider channel width, or smaller Manning's coefficient result in less flooded area and larger fluctuations and advanced peaks in simulated discharge.

To reduce the complexity of calibrating multiple parameter coefficients, the parameters were calibrated by keeping four of the six coefficients constant while varying the two remaining coefficients: (1) coefficients of the river width: $x_w$, $y_w$, (2) coefficients of the river bank height: $x_b$, $y_b$, and (3) coefficients of minimum river width and bank height: $z_w$, $z_b$. To test the robustness of the calibrated parameters, the parameters were perturbed by varying two coefficients according to Fig. A1 while keeping the other four coefficients equal to the previously calibrated coefficients by Mateo et al., (2013, 2014). In total, 24 parameter sets are tested (parameter sets W5, B5, and WB5 are similar to the calibrated parameters). To verify the stability of the calibration with varying spatial scales, the numerical experiment was carried out in two resolutions: (1) 10km (base simulation), and (2) 4km.

The parameter sets were evaluated based on the following metrics: Pearson correlation coefficient, NSE, PBIAS, difference in the magnitude of peak discharge, difference in the timing of the peak discharge, and fit of flood extent (for the entire basin as compared with satellite images on October 25, 2011). Thresholds (values in parentheses) were set to determine the acceptable parameter sets: high correlation coefficient (above 60%), high NSE (above 60%), low PBIAS (absolute value below 10%), low error in peak magnitude (less than 10%), low error in date to peaking (less than 10 days), and high flood extent statistics. The 'optimum' parameter set was determined by (1) calculating the performance of each parameter set in each metric, (2) screening out parameter sets that do not satisfy all of the thresholds used for evaluation, (3) ranking the remaining parameter sets in each metric, (4) giving equal weight to each metric to obtain the simple average rank of the parameter sets, and (5) getting the highest ranking (low rank value) parameter set.

Table A1 summarizes the results of the ranking for parameter sets W1 to W9 and B1 to B9 for the two spatial resolutions. It was found that the model is not sensitive to changes in coefficients $z_w$ and $z_b$ (similar to the results of sensitivity and calibration tests by Mateo et al (2013, 2014)) when applied to the Chao Phraya River Basin at the two spatial resolutions; therefore, the results for parameter sets WB1 to WB9 are not shown in the table for simplicity. Table A1 shows that alternative parameter sets which give comparable results with the calibrated parameters exist (e.g. B7 or W3 as shown in Table A1). However, it should be noted that those parameter sets produce river widths and river bank heights that are within 10% difference in size as compared with those produced using the calibrated parameter set. Hence, the results confirm that the 'optimum' parameter set do not significantly change with spatial scale. This is most likely due to the use of the same runoff data, $R_{up}$, to calculate the sub-grid parameterization of the topography (river width and river bank height); because upstream areas do not greatly vary with spatial scale, the sub-grid characteristics of the main river channels do not change as well.

**Acknowledgements**

This study was supported by the Japan Society for the Promotion of Science KAKENHI (16H06291) and the Integrated study on hydroMeteorological Prediction and Adaptation to Climate change in Thailand (IMPAC-T) Project through the Science and Technology Research Partnership for Sustainable Development (SATREPS). We are thankful to all the people, especially the Thai government officials, who have provided the hydrologic data which were used to calibrate and validate the models in

this study. We are also grateful to the two anonymous referees who have provided constructive comments which led to improvements in the paper. The main author is also grateful to her thesis panelists, Toshio Koike, Hiroaki Furumai, Yukiko Hirabayashi, and Naota Hanasaki, who have critically examined, and thereby significantly contributed to the scientific merit of this study, and Commonwealth Scientific and Industrial Research Organisation (CSIRO) for providing support.

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

**Tables**

**Table 1. CaMa-Flood sub-grid parameters in Fig. 1 (based from Yamazaki et al., 2011)**

| Symbol | Parameter or Variable Meaning | Unit |
|--------|-------------------------------|------|
| *Parameters* | | |
| $A_c$ | unit catchment area | $m^2$ |
| $B$ | bank height | $m$ |
| $L$ | channel length | $m$ |
| $W$ | channel width | $m$ |
| $Z$ | surface altitude | $m$ |
| *Variables* | | |
| $A_f$ | flooded area | $m^2$ |
| $D_r$ | river water depth | $m$ |
| $D_f$ | floodplain water depth | $m$ |
| $S$ | total water storage, $S_r + S_f$ | $m^3$ |
| $S_r$ | river channel water storage | $m^3$ |
| $S_f$ | Floodplain water storage | $m^3$ |

**Table 2. Statistical information on the sub-grid characteristics of river channels.**

| River Characteristic | Map Resolution | | | | | | Trend |
|---|---|---|---|---|---|---|---|
| | 10k | 8k | 6k | 4k | 2k | 1k | |
| Number of land unit catchments | 4534 | 7068 | 12541 | 28178 | 112447 | 449285 | |
| Ratio (Wide river) | 0.15 | 0.12 | 0.09 | 0.07 | 0.03 | 0.02 | |
| Ratio (Narrow river) | 0.85 | 0.88 | 0.91 | 0.93 | 0.97 | 0.98 | |
| Ratio (Deep river) | 0.19 | 0.15 | 0.12 | 0.08 | 0.04 | 0.02 | |
| Ratio (Shallow river) | 0.81 | 0.85 | 0.88 | 0.92 | 0.96 | 0.98 | |
| Max. width (m) | 326.62 | 326.62 | 326.62 | 326.62 | 326.62 | 326.62 | |
| Mean width (Wide river) | 167.33 | 166.77 | 167.22 | 166.94 | 167.31 | 167.32 | |
| Mean width (Narrow river) | 36.52 | 32.93 | 28.62 | 23.47 | 16.44 | 11.26 | |
| Max. height (m) | 4.96 | 4.96 | 4.96 | 4.96 | 4.96 | 4.96 | |
| Mean height (Deep river) | 2.96 | 2.95 | 2.95 | 2.96 | 2.95 | 2.95 | |
| Mean height (Shallow river) | 1.09 | 1.01 | 0.92 | 0.81 | 0.63 | 0.49 | |

* : increasing    : decreasing    : no significant change

**Table A1. Summary of mean ranks for each parameter set at 10km and 4km spatial resolution.** Boxes in gray indicate that the parameter set had been screened out in at least one of the criterion used for evaluation.

| Parameter Set | Mean Rank | | | |
|---|---|---|---|---|
| | B (10km) | W (10km) | B (4km) | W (4km) |
| 1 | $P_t$ | $P_m, P_t$ | $B, P_m, P_t$ | $P_m, P_t$ |
| 2 | $P_t$ | $P_t$ | $P_m, P_t$ | $P_m, P_t$ |
| 3 | $P_t$ | 2 | $P_t$ | 1 |
| 4 | $P_t$ | $P_t$ | $P_t$ | $P_m, P_t$ |
| 5 | 2 | 1 | 1 | 2 |
| 6 | 4 | $P_m$ | 4 | $P_m$ |
| 7 | 1 | $P_t$ | 2 | $P_m, P_t$ |
| 8 | 3 | 3 | 3 | 3 |
| 9 | $N, P_m, P_t$ | $N, P_m, P_t$ | $N, P_m, P_t$ | $N, P_m, P_t$ |

Reason for screening out the parameter set from the ranking: N - low NSE, B - high bias, $P_m$ - high difference in the magnitude of peak discharge, $P_t$ - high difference in the timing of peak discharge

**Figures**

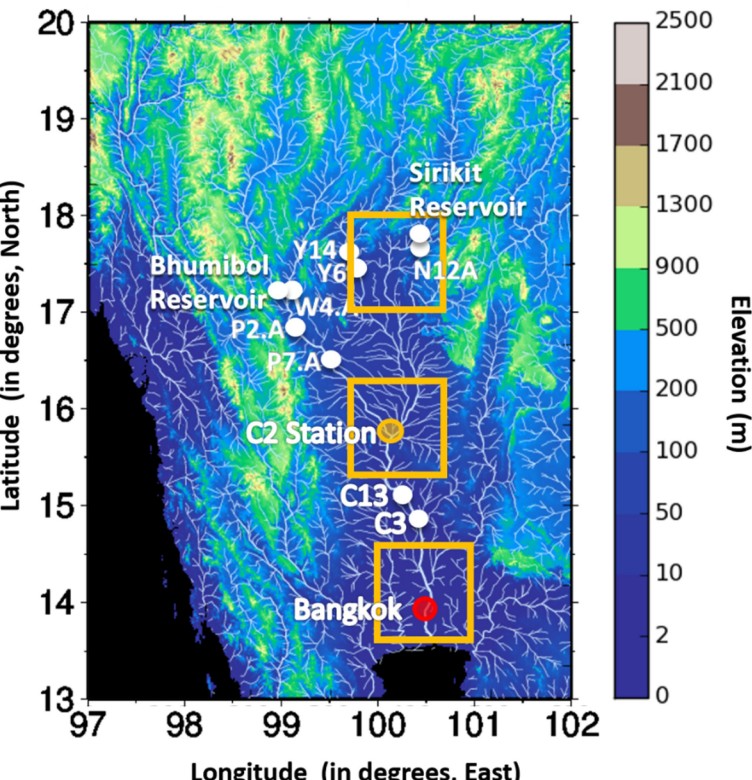

5 **Figure 1. The Chao Phraya River Basin.** Model domain, elevation map, river network, and validation stations. The orange dot marks the location of the station used for calibration. White dots mark the stations used for validation. The red dot marks the capital of Thailand, Bangkok. Areas in orange squares are zoomed in for analysis of results.

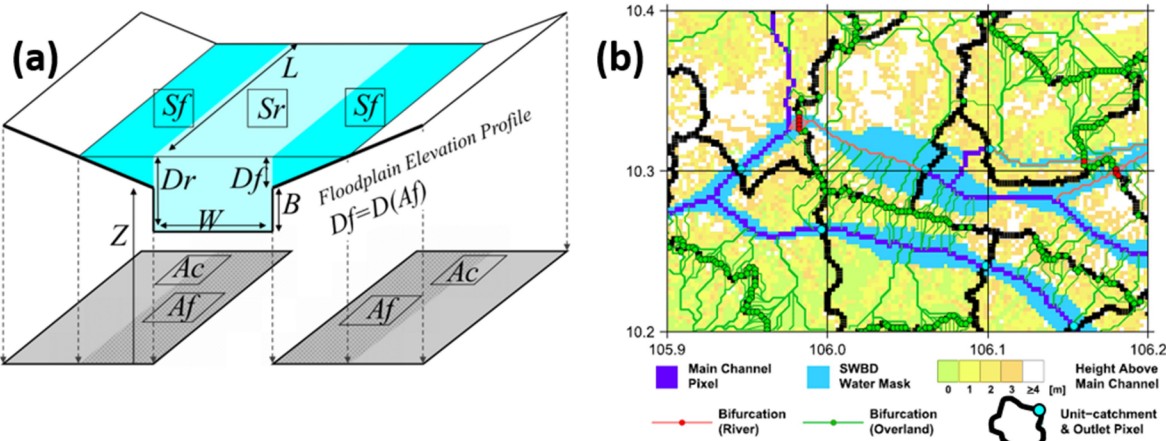

**Figure 2. Schematic diagram of CaMa-Flood.** (a) River channel and floodplain sub-grid parameters (modified from Yamazaki et al., 2011). Please refer to Table 1 for definition of variables. (b) Sub-grid topography and *MDC* channels. Overland pathways are represented by the green channels while river pathways are represented by the red channels (modified from Yamazaki et al., 2014b).

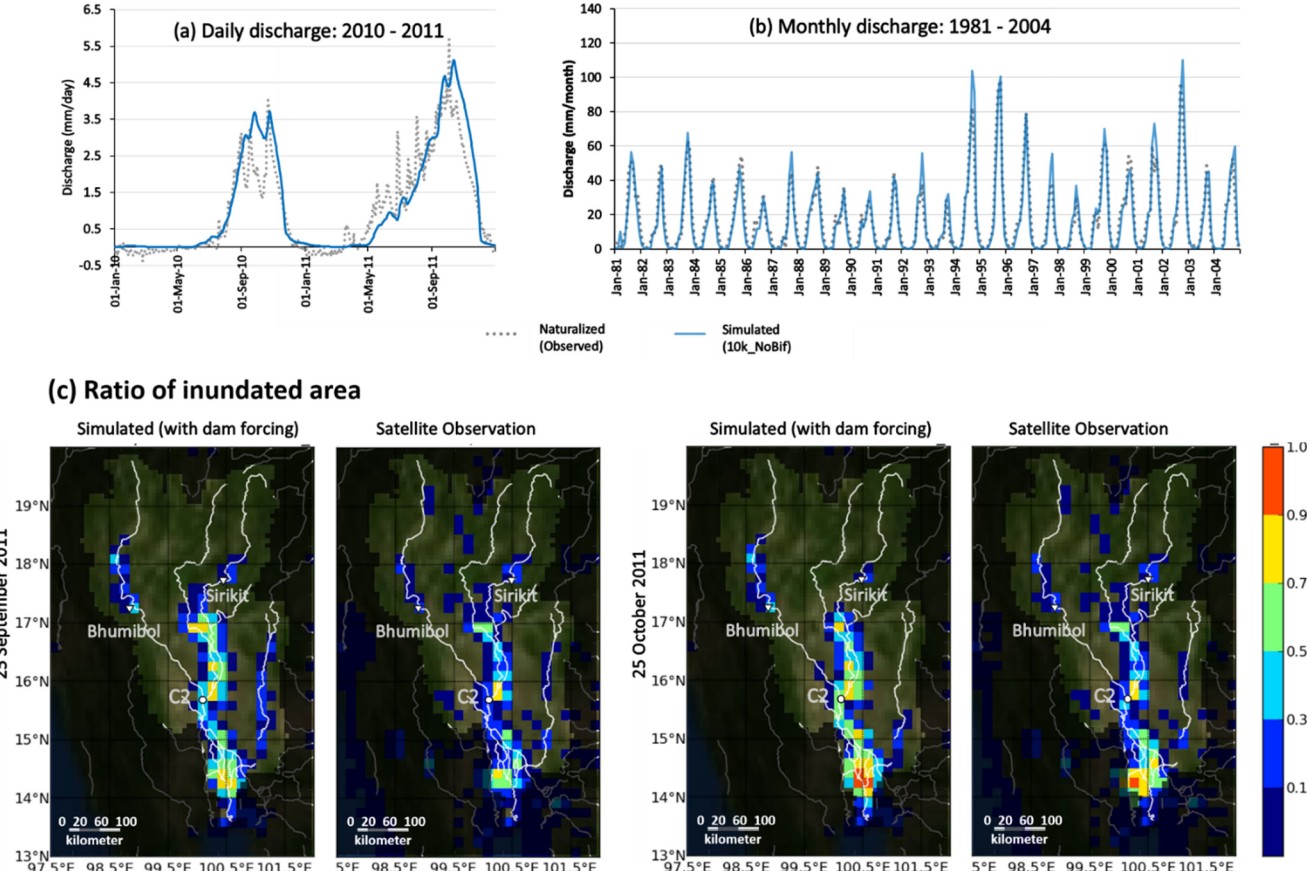

**Figure 3. Validation of *SDC* simulation at 10k resolution.** The simulated (a) daily discharge for the years 2010-2011 and the (b) monthly discharge from 1981-2004 were compared with the naturalized observed discharge. The (c) ratio of inundated area in September 25, 2011 and October 25, 2011 were compared with satellite data. Dam forcing indicates the simulations constrained using actual dam outflows as boundary conditions. All figures were modified from Mateo et al. (2014).

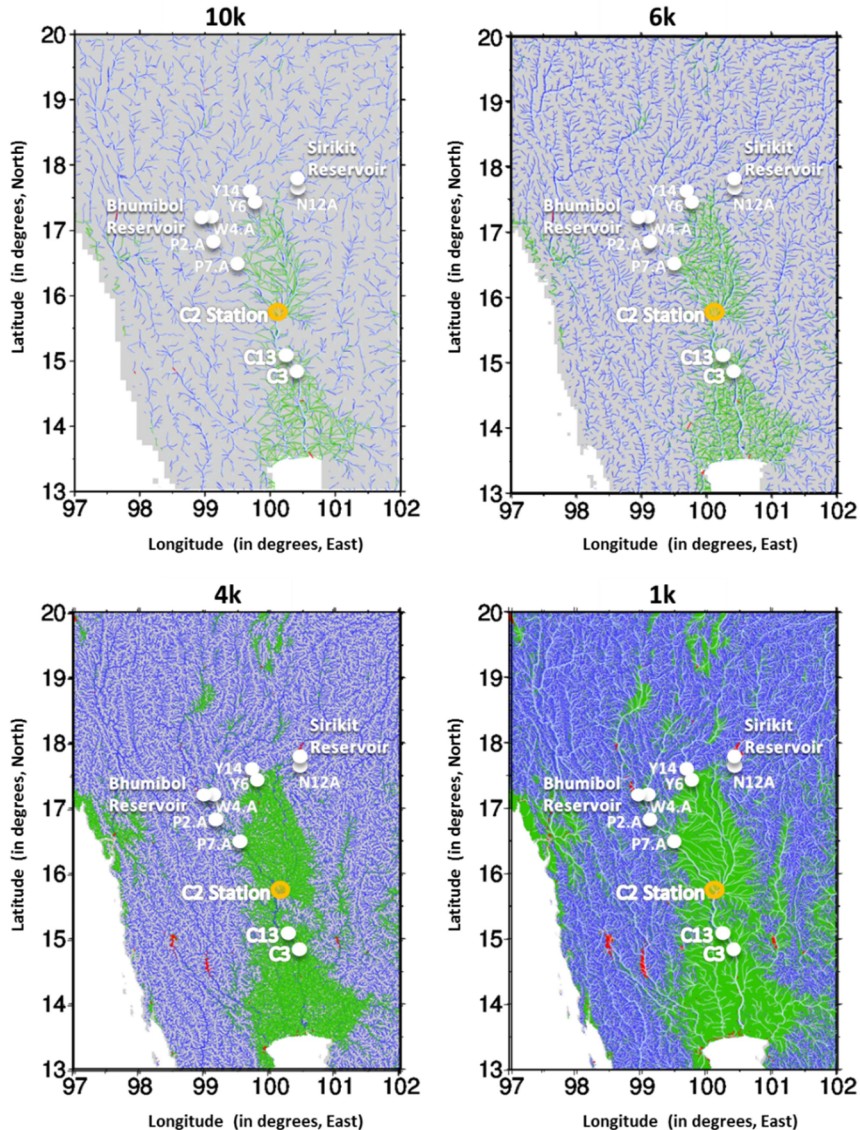

**Figure 4. Change in the density of *MDC* pathways with spatial resolution.** The density of river network and MDC pathways increase with increasing spatial resolution. Gauge stations upstream of C2 Station (Y14, Y6, N12A, W4A, P2A, and P7A) are located in zones with low density of MDC pathways. Gauge stations C2, C13, and C3 are located in low-lying and floodplain areas which have a high density of MDC pathways.

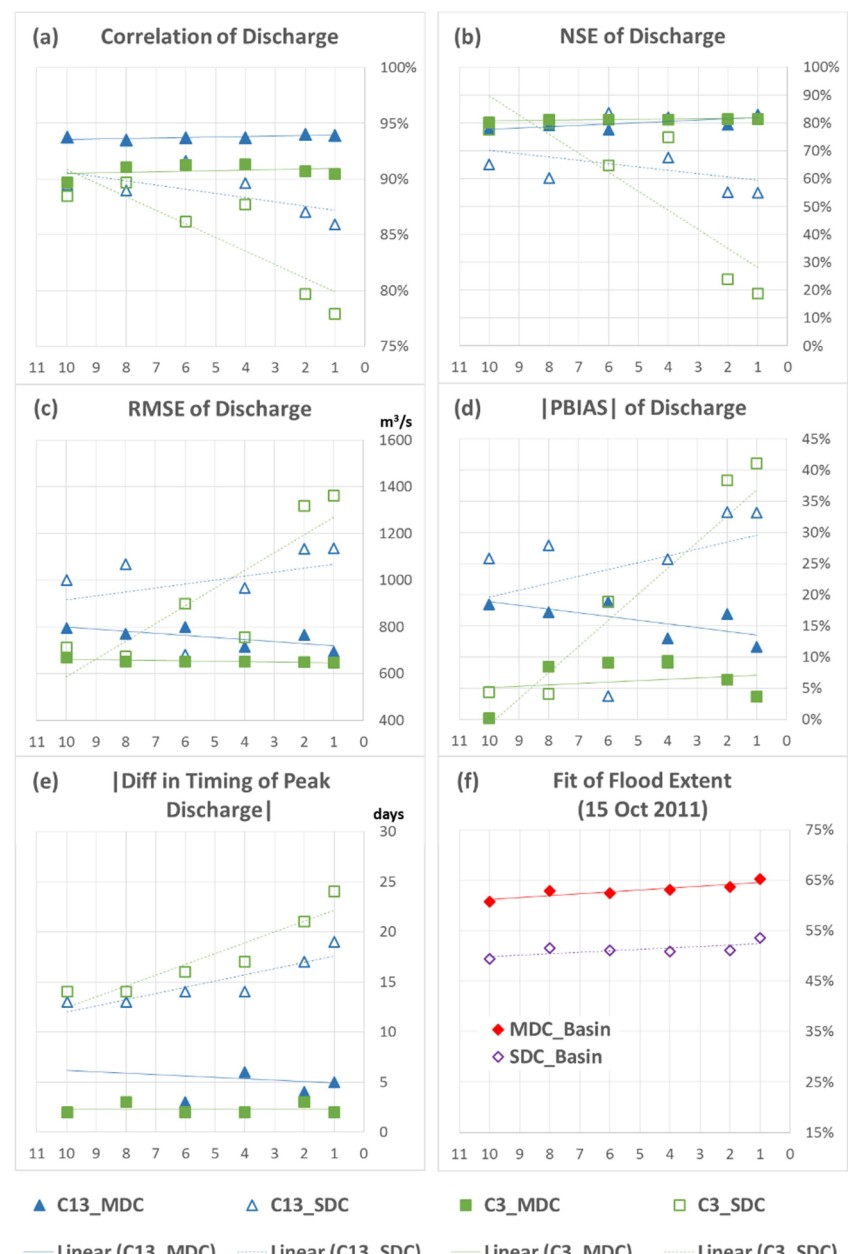

**Figure 5. Model efficiency at varying spatial resolution and varying flow connectivity schemes.** The statistics for discharge (Correlation, RMSE, Percent Bias, NSE Coefficient, and difference in peak timing) were calculated and compared with the naturalized discharge at corresponding gauge stations. The fit of flood extent was calculated for the entire basin.

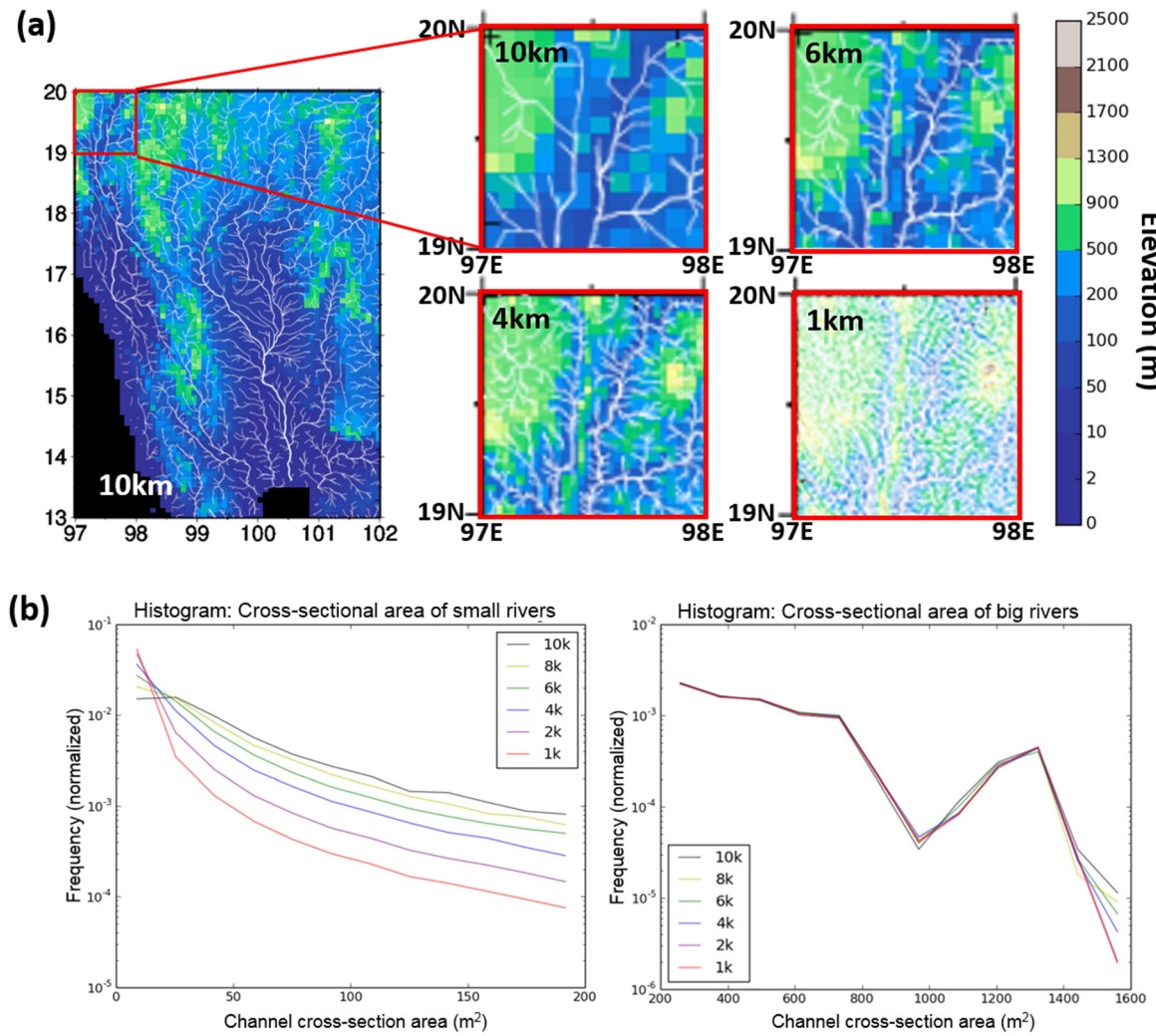

**Figure 6. Impacts of increasing spatial resolution to the topographic and river channel characteristics in the model.** (a) The elevation and sub-grid representation of rivers (white lines) become increasingly detailed with finer spatial resolution. (b) The distribution of cross-sectional area of small (cross-section area < 200m$^2$) and big (cross-section area >= 200m$^2$) rivers in varying spatial resolutions; the distribution of small rivers significantly change with varying spatial resolution.

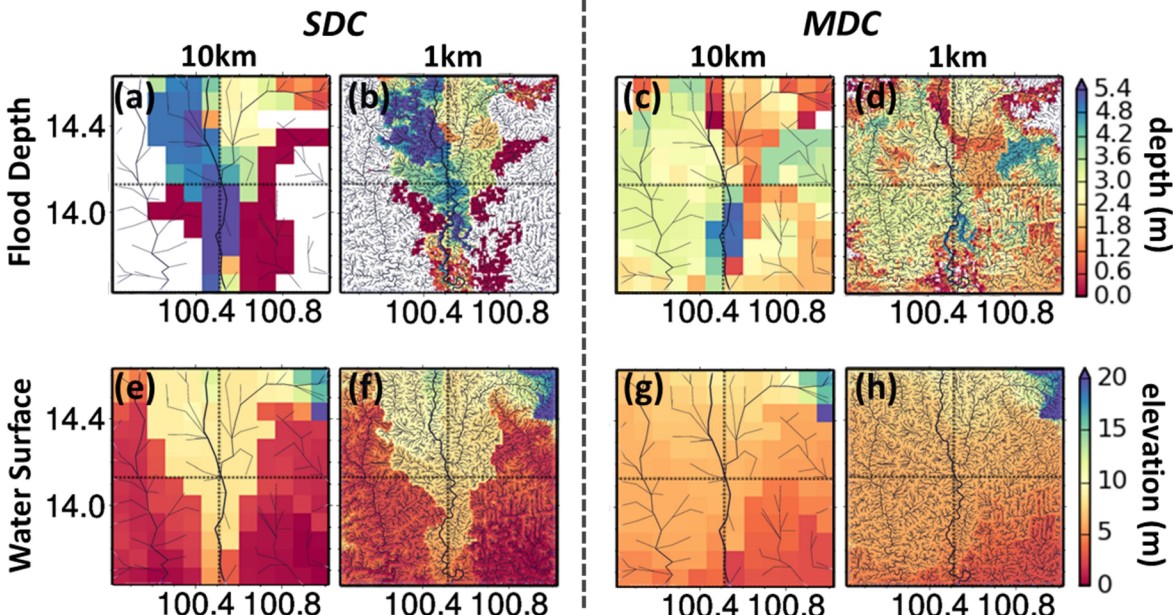

**Figure 7. Impacts of spatial resolution and representation of *MDC* on simulated flood depth and water surface.** (a – d) Flood depth and (e – h) water surface at the lower part of the basin. The first two columns show the results for *SDC* simulations in 10km and 1km resolutions while the last two columns show the results for *MDC* simulations.

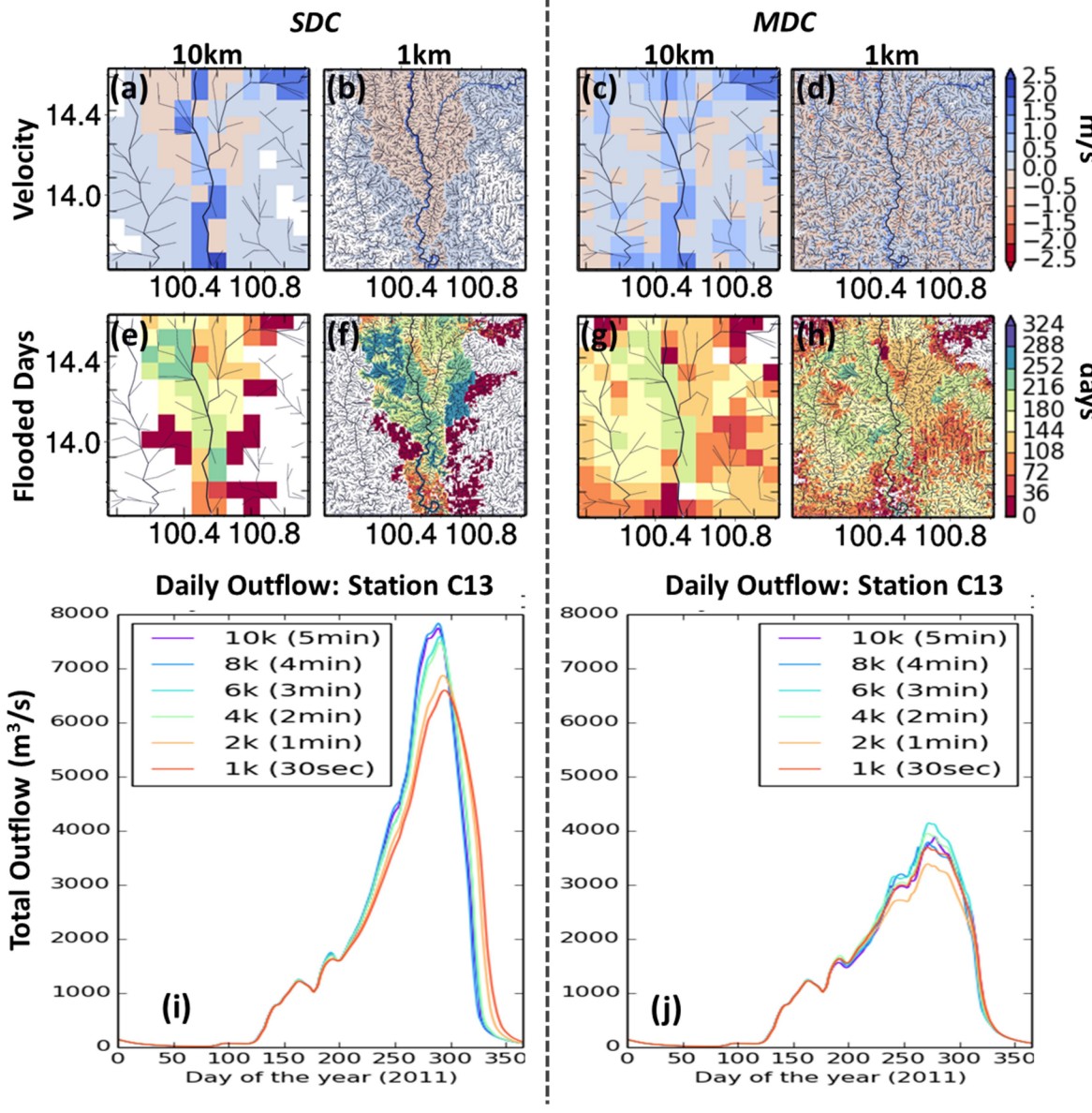

**Figure 8. Impacts of spatial resolution and representation of *MDC* to (a – d) flow velocity, (e – h) number of days of flooding in a year, and (i – j) 2011 daily outflows (river + floodplain flows) at C13 Station.** Figures a to h show simulation results at the lower part of the basin.

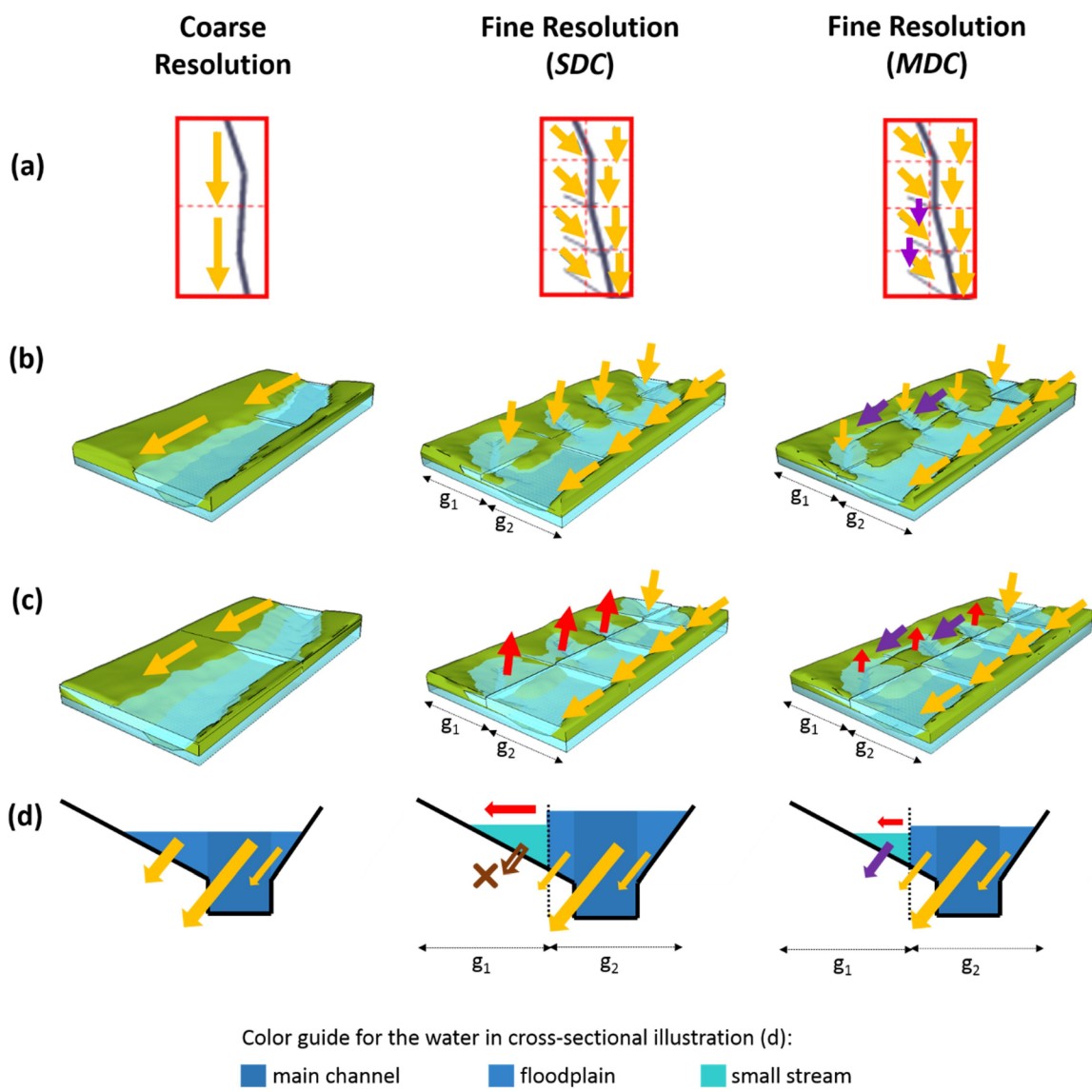

**Figure 9. Schematic diagram of the initial distribution of water in coarse and fine resolution simulations.** (a) Top view of the river flow network. (b) Isometric 3D representation of the section of the river. Here, to visualize the small rivers and main channels, vertical features of the floodplain are illustrated with exaggeration. Arrows depict flow direction during non-flood conditions. (c) Isometric and (d) cross-sectional illustration of the flow direction in the river network during flooding. Yellow orange arrows indicate the flow of water to the downstream, purple arrows indicate flow in *MDC* pathways, red arrows indicate backflows, and brown arrows succeeded by "X" marks indicate no flow connectivity in that direction.

$z_w$=3.0; $x_b$=0.70; $y_b$=0.23; $z_b$=0.20    $z_b$=0.20; $x_w$=16.6; $y_w$=0.35; $z_w$=3.0    $x_w$=16.6; $y_w$=0.35; $x_b$=0.70; $y_b$=0.23

| | $y_w$ = 0.30 | $y_w$ = 0.35 | $y_w$ = 0.40 |
|---|---|---|---|
| $x_w$ = 12 | W1 | W2 | W3 |
| $x_w$ = 16 | W4 | W5 | W6 |
| $x_w$ = 20 | W7 | W8 | W9 |

| | $y_b$ = 0.18 | $y_b$ = 0.23 | $y_b$ = 0.28 |
|---|---|---|---|
| $x_b$ = 0.4 | B1 | B2 | B3 |
| $x_b$ = 0.7 | B4 | B5 | B6 |
| $x_b$ = 1.0 | B7 | B8 | B9 |

| | $z_w$ = 1.0 | $z_w$ = 3.0 | $z_w$ = 5.0 |
|---|---|---|---|
| $z_b$ = 0.1 | WB1 | WB2 | WB3 |
| $z_b$ = 0.2 | WB4 | WB5 | WB6 |
| $z_b$ = 0.5 | WB7 | WB8 | WB9 |

**Figure A1. Parameter sets used in checking the transferability of calibration in 10km and 4km spatial resolutions.**