# Peer review of "Impacts of spatial resolution and representation of flow connectivity on large-scale simulation of floods"

_Hydrology and Earth System Sciences, 2016_

## Referee Comment (RC1) · Anonymous Referee #1 · 3 Apr 2017

This paper reports on the application of a new model structure in the CaMa-flood Global River Model (GRM) to the simulation of the Thailand 2011 flood. The new model structure introduces extra connectivity in CaMa-flood to allow better simulation of river flows in cases where the simple assumption of a single downstream drainage path, which is the common drainage structure used in GRMs, does not apply (for example in deltas where there is river bifurcation). The need for improved connectivity and better river representation in GRMs has been identified as an important area of research, therefore this paper is a very timely and important contribution to the evolution of GRMs in general.

The paper is mostly well written and clear except for the few typos listed below. Figures

are also of good quality and informative. Analysis on the whole seems thorough and appropriate. Where I think there are some issues is in the interpretation of the findings and some claims which maybe a bit overconfident, given the evidence. I feel the paper should be published after some minor additional discussion to cover these conceptual issues.

Conceptual issues:

(1) In the abstract (line 24), it is stated that the findings are universal. However I do not think evidence is provided to back up this claim. I agree that, in theory they should be, but without wider testing or evidence, this is an over bold claim. I would suggest, either rewording this to "findings should be universal" or providing a better argument as to why the authors consider them universal, for example through a clear geomorphological explanation as to why the findings here equally apply elsewhere.

(2) The link between the real river system and the model representation of that system is not very well explained or developed. While GRMs will obviously necessitate approximations, as they strive for higher resolution representation and improved physical representation, it is important that these additions/improvements are providing the right results for the right reasons. For example, interconnecting flood flows across floodplains occurs through two main processes, diffusive overland flow and channelized flow. Many of the smaller channels are not explicitly present in the DEMs used in the models and even the finer resolution reach-scale hydrodynamic models struggle to include this complexity. The method to improve connectivity in the CaMa-flood model seems to be a diffusive flow method between cells and will not explicitly capture smaller floodplain channels. Any smaller channels present on the floodplain in the CaMa-flood model will be DEM drainage paths rather than real observed channels, so it is not clear how the extra model connectivity relates to the real river systems. The reason this is important is that there is a danger of introducing too much connectivity. This will have the effect of improving the models ability to capture a flood but at the expense of over prediction. I would suggest that the representation of real observed flood connectivity

is covered in more detail in the discussion (there are plenty of good papers on this topic – including on the Chao Phraya).

Minor typos and suggested edits:

(1) Page 1, line 14: "simplified representations" instead of "simplified representation"

(2) Page 2, line 20: "simplified representations" instead of "simplified representation"

(3) Page 2, line 31: "flood hazard maps for 1 x 1 km grids" instead "flood hazard maps at 1 x 1 km grids"

(4) Page 3, line 29: "model has been" instead of " model have been"

(5) Page 4, line 18: "This paper assessed"….. papers don't assess, people do. Perhaps "In this paper we assess". Also "flood in the Chao Phraya River Basin" instead of "flood in Chao Phraya River Basin"

(6) Page 4, line 30: "flood in recent history" instead of "flood in history" or define the time period of history.

(7) Page 5, line 5: "based on fine-resolution" instead of "based from fine-resolution"

(8) Page 5, line 8: "DEM" instead of "DEMs"

(9) Page 6, line 27: "The simulation" instead of "Simulation" and "The calculation" instead of "Calculation"

(10) Page 7, line 16: "simulation for different" instead of "simulation at different" and "spatial resolutions" instead of "spatial resolution"

(11) Page 7, line 21: "explain" instead of "explicate"?

(12) Page 7, line 22: "dynamics are discussed" instead of "dynamics is discussed"

(13) Page 7, line 24: there is a missing close bracket ")"

(14) Page 8, line 10: first mention of dams on the system. Might be good to add a bit

of info on the dams and their role in section 2

(15) Page 8, line 16: "relatively high" instead of "relatively higher" and also "govern more than other flood" instead of "govern over other flood"

(16) Page 9, line 12: "process resulted in an increase" instead of "process resulted to an increase"

(17) Page 9, line 24: "do not have explicit upstream-downstream channel relationships" instead of "do not have upstream-downstream relationships"

(18) Page 9, line 29: "shows an increasing number of unit-catchments with backflow" instead of "shows increasing number of unit-catchments in backflow"

(19) Page 10, line 13: "results in more rivers" instead of "results for more rivers"

(20) Page 14, line 3: "result in low errors" instead of "result to low errors"

---

## Referee Comment (RC2) · Anonymous Referee #2 · 4 Apr 2017

This paper explores the impact of spatial resolution and flow connectivity on the prediction of flood inundation at basin scales. The aim is to explore whether increased spatial resolution improves the prediction of flood levels by global-scale river models. The paper tests this at one site (Chao Phraya River Basin, Thailand), and while the results show that increased spatial resolution can lead to improvements in the prediction of flood levels, this requires consideration of multiple connections in regions like river deltas. Overall, the paper presents an interesting result which would be of value to the community, and should be published after the authors have considered the comments below.

[Figure]

Specific comments

Only having one site means that extrapolating the result to global-scale models is more difficult. The authors could argue that the problem with single downstream connectivity in a model will impact many (if not most) floodplains globally, on the basis of 1 study site it is difficult to evaluate how common the problem will be. To reach this conclusion, a SDC and an MDC model need to be compared globally to see exactly how significant the conversion to an MDC model would be. The problem with doing this is that, as the authors state, calibration of the parameters of the MDC model is a tedious process. This will limit global applications of such a model. Ideally what is needed as the next step is a means of more easily (i.e. automatically) calibrating the parameter values. This could be by developing a way of doing this, or by reformulating the model to enable this to be done. This doesn't subtract from the significance of this paper. The paper clearly shows that in the case considered, the MDC approach is necessary in order to improve model predictions.

The authors mention that more detailed hydrodynamic models are still needed to model the behaviour of floods at smaller scales (line 21, page 11). The authors might want to comment on the possibility of the development of a hybrid approach, where the GRM result is used as a starting point (either boundary or initial conditions) of a finer scale model – possible at the end of that paragraph.

It would be good to see a sensitivity analysis of these model. How important are the sub-grid channel parameters in terms of the modelled output?

Lines 19 to 21, page 6: The authors might need to clarify this sentence a little. Does this mean that the CaMa-Flood values were adjusted to ensure that the catchment scale output matched the gridded runoff within the catchment? If so, how significant was this adjustment? Line 14, page 7: What are the confidence bounds in the values for Manning's coefficient? How do these confidence bounds impact on the model output? Is uncertainty in these values captured by other parameters?

Technical corrections

There are a significant number of typographical and grammatical errors in the paper (see the comments from the first reviewer for a fairly comprehensive list of these).

Line 28, page 3: I would suggestion "CaMa-Flood is currently the only GRM . . ."

Line 7, page 4: I think "can be answered" is a bit strong here (see comment above). Actually, all that has been done is investigate these questions from the point of view of a case study.

Line 14, page 5: I think "Yamazaki et al., 2014b" is the wrong citation here. This makes it seem that the "local inertial equation" was first presented in this paper. A couple of lines later, the authors refer to a 2010 paper that talks about it. Options are to remove the citation (already given elsewhere in the paper), or give a more appropriate citation to the "local inertial equation".

Line 16, page 5: either "represent the backwater effect" or "represents backwater effects"

Line 27, page 5: either "the" or "a" before "bifurcation scheme"

Line 2, page 8: Are the objective functions evaluated at a site-by-site level, or are the aggregated across a number of sites. Judging by the rest of the text, it is done at a site level. If so, then there is a simple 1-to-1 relationship between NSE and RMSE, so using both would not give any additional information.

Line 3, page 8: Here the authors use "difference in discharge peak timing". Elsewhere (e.g. line 23, page 8; line 33, page 9) they use "delay". The authors should be consistent. In my view "difference" would be the better term to use as it is the difference in the delay in the peak that is being measured.

Line 33, page 9: change "time to peaking" to "time to peak"

Line 4 to 7, page 12: There is an apparent contradiction in these two sentences. The

first says that the calibration of the sub-grid channel parameters is tedious – implying that this was a time consuming process. Later the authors state that the calibrated parameter values were found to be consistent and transferable across spatial resolutions – implying that calibration wasn't that difficult.

Line 2, page 13: change "it significantly declined" to either "this significantly declined" or "it declined significantly"

---

## Author Comment (AC1) · 2 May 2017

Please refer to the document "Response to the comments of Referee 1.pdf" in the attached supplement for our response to all the comments of Referee #1.

We have also included the following documents in the attachment:

1) Revised manuscript: HESSD_Mateo_etal_2017_Impacts_of_spatial_resolution_and_ representation_of_flow_connectivity_revised_manuscript.pdf

2) Revised manuscript with all changes tracked: HESSD_Mateo_etal_2017_Impacts_of_spatial_resolution_and_representation_ of_flow_connectivity_tracked_changes.pdf

3) Response to the comments of both referees: Response to Referees.pdf

Please also note the supplement to this comment:
http://www.hydrol-earth-syst-sci-discuss.net/hess-2016-620/hess-2016-620-AC1-
supplement.zip

---

## Author Response (AR1)

**Response to referees' comments on "Impacts of spatial resolution and representation of flow connectivity on large-scale simulation of floods" by Cherry May R. Mateo et al.**

5    Note: This document provides authors' responses to the comments of Referee #1 and Referee #2. We have formatted the comments made by the referees in italic and black-colored fonts and our response in upright and blue-colored fonts. To distinguish the modifications made in the manuscript from our response to the referees' comments, modifications are formatted in a different font type (Times New Roman). The page and line numbers within brackets or parentheses indicate the location of the

10   modifications in the revised manuscript. The revised manuscript with all revisions tracked is appended at the end of this document.

**Response to the comments of Referee #1**

*This paper reports on the application of a new model structure in the CaMa-flood Global River Model*

15   *(GRM) to the simulation of the Thailand 2011 flood. The new model structure introduces extra connectivity in CaMa-flood to allow better simulation of river flows in cases where the simple assumption of a single downstream drainage path, which is the common drainage structure used in GRMs, does not apply (for example in deltas where there is river bifurcation). The need for improved connectivity and better river representation in GRMs has been identified as an important area of*

20   *research, therefore this paper is a very timely and important contribution to the evolution of GRMs in general.*

*The paper is mostly well written and clear except for the few typos listed below. Figures are also of good quality and informative. Analysis on the whole seems thorough and appropriate. Where I think there are some issues is in the interpretation of the findings and some claims which maybe a bit*

25   *overconfident, given the evidence. I feel the paper should be published after some minor additional discussion to cover these conceptual issues.*

We thank the referee for recognizing the timeliness and importance of our work. We greatly appreciate the time and effort that the referee have spent in reviewing the manuscript. The suggestions made by the referee are very helpful in improving the discussions in the revised

30   manuscript. We have corrected the typos pointed out by the referee. We have also modified the parts which readers may find to be overconfident. Where necessary, we have included a brief discussion to support our claims. Please refer to the revised manuscript for the changes made.

*Conceptual issues: (1) In the abstract (line 24), it is stated that the findings are universal. However I do*

35   *not think evidence is provided to back up this claim. I agree that, in theory they should be, but without wider testing or evidence, this is an over bold claim. I would suggest, either rewording this to "findings should be universal" or providing a better argument as to why the authors consider them universal, for example through a clear geomorphological explanation as to why the findings here equally apply elsewhere.*

40   We thank the referee for pointing this out. We do recognize the point made by the referee that while the findings are universal in theory, the paper shows the validity of the theory in only one region.

Further tests may be needed to support the claim. Hence, we have modified the statement in the abstract according to the referee's suggestion:

"While a regional-scale flood was chosen as a test case, these findings should be universal and may have significant impacts on large- to global-scale simulations especially in regions where mega deltas exist." [page 1, lines 25-26]

We have also toned down a statement in the introduction from "With this verification exercise, two fundamental questions with regards to large- to global-scale simulation of floods can be answered…" to "With this verification exercise, we attempt to answer two fundamental questions with regards to large- to global-scale simulation of floods…" [page 4, lines 11-12]

We argue that because the mechanism behind our findings is physically-based, the findings should theoretically be applicable globally, but the impacts will be more evident in river mega deltas. The reasons for the decline in the capability of GRMs using *SDC* scheme to simulate flooding in floodplains and mega deltas in finer spatial resolution have been discussed in 6.1. We have added a few lines in the section to further improve the discussion (please see page 10, line 22 to page 11, line 11 of the revised manuscript).

*(2) The link between the real river system and the model representation of that system is not very well explained or developed. While GRMs will obviously necessitate approximations, as they strive for higher resolution representation and improved physical representation, it is important that these additions/improvements are providing the right results for the right reasons. For example, interconnecting flood flows across floodplains occurs through two main processes, diffusive overland flow and channelized flow. Many of the smaller channels are not explicitly present in the DEMs used in the models and even the finer resolution reach-scale hydrodynamic models struggle to include this complexity. The method to improve connectivity in the CaMa-flood model seems to be a diffusive flow method between cells and will not explicitly capture smaller floodplain channels. Any smaller channels present on the floodplain in the CaMa-flood model will be DEM drainage paths rather than real observed channels, so it is not clear how the extra model connectivity relates to the real river systems. The reason this is important is that there is a danger of introducing too much connectivity. This will have the effect of improving the models ability to capture a flood but at the expense of over prediction. I would suggest that the representation of real observed flood connectivity is covered in more detail in the discussion (there are plenty of good papers on this topic – including on the Chao Phraya).*

Thank you for raising this important issue. We generally agree with the referee on the points made. We have added a few lines to describe how the MDC channels are derived in the model in section 3.1:

"The algorithm for extracting bifurcation channels is described in detail in Yamazaki et al. (2014b) and will only be described briefly in this paper. Using data from HydroSHEDS and SRTM3, the algorithm searches for possible flow pathways which cross unit-catchment boundaries. A "bifurcation threshold height" above the main channel of each unit-catchment is set for computational efficiency. The algorithm searches for pixels (grid cells in the SRTM3 DEM) which are at unit-catchment boundaries and are at an elevation lower than that of the bifurcation threshold. The pixel is identified as a valid bifurcation point if its elevation is higher than that of an adjacent pixel which is located in another unit-catchment. Using HydroSHEDS flow directions, a bifurcation channel is defined as the pathway from each bifurcation point to the main channel pixels of its upstream and downstream unit-catchments. Bifurcation channels in floodplains are represented by overland pathways, while those with persistent bifurcated flow are represented by river pathways. Persistent bifurcated flow is detected using the SRTM Water Body Data (SWBD) water mask (NASA/NGA, 2003)." [page 6, lines 9-18]

We agree with the referee that the issue of representing connectivity (as well as other processes that influence flow pathways) should be discussed more in the paper. To address this issue, we have added a paragraph in section 6.3 (caveats and future works):

"While the use of the *MDC* scheme in CaMa-Flood resulted in improvements in the simulation of flood dynamics in large floodplains, it should be noted that uncertainties remain in the representation of *MDC* pathways in the model. The *MDC* pathways in the model may not necessarily correspond to or explicitly represent the actual flow pathways, especially the small channels (e.g. small artificial canals) in the river basin. Small *MDC* channels in the model which are not covered by the SWBD water mask are currently represented as overland pathways. As channel bathymetry is not considered in overland pathways, this assumption may lead to the underestimation of flows in small *MDC* channels in the model (Yamazaki et al., 2014b). The accurate representation of *MDC* pathways in the model still depends on the resolution and accuracy of the DEM used (3-arcsecond or ~90m SRTM3 DEM by Farr et al., 2007 in this study). The explicit representation of small artificial channels with widths which are narrower than the grid resolution of the DEM used and other small scale flow connectivity between rivers and floodplains is still difficult to achieve in large scale simulations. A finer resolution DEM or a tool for extracting or deriving smaller channels from open street maps will be very helpful in improving the accuracy of the representation of *MDC* pathways in the model. The use of new data-driven approaches to derive flood-mediated *MDC* pathways and connectivity (e.g. progressive nearest neighbour search or progressive iterative nearest neighbour search by Zhao et al., 2017) may also be explored in the future. It should also be noted that the changes in channel network (by sedimentation, subsidence, and other geological processes, or by levee breaks, water diversion, and other anthropogenic impacts) and operation of artificial canals are not represented in the current model. Such natural or anthropogenic influences which add to the complexities in real flow pathways have significant impacts on the connectivity and flood dynamics in floodplains (Syvitski et al., 2005; Alsdorf et al., 2007; Schumann et al., 2011; Trigg et al., 2013). However, even catchment scale hydrodynamic models implemented in fine spatial resolution have difficulties in representing such complex processes. The representation of such complexities will require the integration of more detailed models and data (e.g. landscape, sedimentation, anthropogenic, etc.) with flood models. " [page 13, line 17- page 14, line 3]

*Minor typos and suggested edits:*

*(1) Page 1, line 14: "simplified representations" instead of "simplified representation"*

*(2) Page 2, line 20: "simplified representations" instead of "simplified representation"*

*(3) Page 2, line 31: "flood hazard maps for 1 x 1 km grids" instead "flood hazard maps at 1 x 1 km grids"*

*(4) Page 3, line 29: "model has been" instead of " model have been"*

*(5) Page 4, line 18: "This paper assessed". . ... papers don't assess, people do. Perhaps "In this paper we assess". Also "flood in the Chao Phraya River Basin" instead of "flood in Chao Phraya River Basin"*

*(7) Page 5, line 5: "based on fine-resolution" instead of "based from fine-resolution"*

*(8) Page 5, line 8: "DEM" instead of "DEMs"*

*(9) Page 6, line 27: "The simulation" instead of "Simulation" and "The calculation" instead of "Calculation"*

*(10) Page 7, line 16: "simulation for different" instead of "simulation at different" and "spatial resolutions" instead of "spatial resolution"*

*(11) Page 7, line 21: "explain" instead of "explicate"?*

*(12) Page 7, line 22: "dynamics are discussed" instead of "dynamics is discussed"*

*(13) Page 7, line 24: there is a missing close bracket ")"*

*(15) Page 8, line 16: "relatively high" instead of "relatively higher" and also "govern more than other flood" instead of "govern over other flood"*

*(16) Page 9, line 12: "process resulted in an increase" instead of "process resulted to an increase"*

*(17) Page 9, line 24: "do not have explicit upstream-downstream channel relationships" instead of "do not have upstream-downstream relationships"*

*(18) Page 9, line 29: "shows an increasing number of unit-catchments with backflow" instead of "shows increasing number of unit-catchments in backflow"*

*(19) Page 10, line 13: "results in more rivers" instead of "results for more rivers"*

*(20) Page 14, line 3: "result in low errors" instead of "result to low errors"*

We thank the referee for pointing out the typos and suggesting how to edit them. We have revised the manuscript and incorporated the suggested edits.

*(6) Page 4, line 30: "flood in recent history" instead of "flood in history" or define the time period of history.*

The EM-DAT records flood in recent history (1900 to present). Thank you for pointing this out. We have changed the wording to "flood in recent history." [page 5, line 3]

*(14) Page 8, line 10: first mention of dams on the system. Might be good to add a bit of info on the dams and their role in section 2*

Thank you for the suggestion. We have added a few lines about the dams in section 2:

"Two huge artificial reservoirs (Bhumibol and Sirikit) and several smaller artificial reservoirs are operational in the Chao Phraya River Basin. In this study, the impacts of reservoir operation on flood flows are removed by using naturalized flows (see Appendix for details) and assessing flood extents on dates when both reservoirs are already full and are assumed to have minimal impact on flooding." [page 5, lines 5-8]

**Response to the comments of Referee #2**

*This paper explores the impact of spatial resolution and flow connectivity on the prediction of flood inundation at basin scales. The aim is to explore whether increased spatial resolution improves the prediction of flood levels by global-scale river models. The paper tests this at one site (Chao Phraya River Basin, Thailand), and while the results show that increased spatial resolution can lead to improvements in the prediction of flood levels, this requires consideration of multiple connections in regions like river deltas. Overall, the paper presents an interesting result which would be of value to the community, and should be published after the authors have considered the comments below.*

We would like to thank Referee #2 for thoroughly reviewing the paper and for giving constructive comments and suggestions which are useful for improving the paper.

*Only having one site means that extrapolating the result to global-scale models is more difficult. The authors could argue that the problem with single downstream connectivity in a model will impact many (if not most) floodplains globally, on the basis of 1 study site it is difficult to evaluate how common the problem will be. To reach this conclusion, a SDC and an MDC model need to be compared globally to see exactly how significant the conversion to an MDC model would be. The problem with doing this is that, as the authors state, calibration of the parameters of the MDC model is a tedious process. This will limit global applications of such a model. Ideally what is needed as the next step is a means of more easily (i.e. automatically) calibrating the parameter values. This could be by developing a way of doing this, or by reformulating the model to enable this to be done. This doesn't subtract from the significance of this paper. The paper clearly shows that in the case considered, the MDC approach is necessary in order to improve model predictions.*

Thank you for recognizing the significance of the paper and for raising the issue about the extrapolation of our results to global simulation. We do recognize that a statement in the abstract about extrapolating our results to global flood simulations may be too bold. Referee #1 raised a similar concern and we have responded by revising a statement in the abstract to:

"While a regional-scale flood was chosen as a test case, these findings should be universal and may have significant impacts on large- to global-scale simulations especially in regions where mega deltas exist." [page 1, lines 25-26]

We argue, however, that because the mechanism behind our findings is physically-based, our findings should theoretically be applicable globally, but the impacts will be more evident in river mega deltas. The reasons for the decline in the capability of GRMs using *SDC* scheme to simulate flooding in floodplains and mega deltas in finer spatial resolution have been discussed in 6.1. We have added a few lines in the section to further improve the discussion (please see page 10, line 22 to page 11, line 11 of the revised manuscript).

As for the issue about calibration, we understand that the model still needs to be validated in other river basins, as we have pointed out in section 6.3:

"More extensive tests in large river basins and at global scale have to be conducted to further validate the model." [page 13, lines 15-16]

While the CaMa-Flood model with *MDC* scheme has not been applied globally, prior to this study, it had been applied to the Mekong River Basin (Yamazaki et al., 2014b) and to the Ganges-Brahmaputra-Meghna Delta (Ikeuchi et al., 2015). In both studies, the *MDC* scheme proved to result in significant improvements in the simulation of flood inundation as compared with the model with *SDC* scheme,

particularly by allowing the flow of water to smaller streams in the deltaic regions. These studies, along with the results presented in this study, suggest the importance of incorporating the *MDC* scheme in simulating floods, particularly in mega deltas.

The SDC version of the model had been applied globally in several studies (e.g. Yamazaki et al., 2011; Hirabayashi et al., 2013). Those studies, however, did not employ any calibration of the parameters due to practicability. Even without calibrating the model, those studies show that the model can be applied globally to allow the analysis of floods from a global perspective. As we have pointed out in the paper, the calibration of the model may limit its application in other large basins but there are several developments that may ease the difficulty of calibration such as the availability of a global width database of large rivers (GWDLR by Yamazaki et al., 2014a).

Thank you for bringing up the issue of automatic calibration. We have included a few lines in the discussion, section 6.3, to recognize the need for a calibration tool:

"One of the caveats of this study is the tedious calibration of the sub-grid channel parameters when applied to regional basins. At the global scale, this can potentially be addressed by the global width database available for large rivers (GWDLR developed by Yamazaki et al., 2014a). A database of channel depths of large rivers, however, do not exist; hence, the parameters characterizing the channel depths in the model may still have to be calibrated. To ease the difficulty of calibration, the development of an automatic calibration tool or a simpler or more efficient parameterization of sub-grid channels (e.g. Neal et al., 2015) may be helpful." [page 13, lines 2-7]

*The authors mention that more detailed hydrodynamic models are still needed to model the behaviour of floods at smaller scales (line 21, page 11). The authors might want to comment on the possibility of the development of a hybrid approach, where the GRM result is used as a starting point (either boundary or initial conditions) of a finer scale model – possible at the end of that paragraph.*

Thank you for the really helpful suggestion. We have added the following at the end of section 6.2:

"To harness the benefits from using GRMs and catchment-scale hydrodynamic models, the development of hybrid approaches, where outputs from CaMa-Flood with *MDC* scheme are used as initial or boundary conditions of catchment-scale hydrodynamic models, may be developed and assessed in the future. Hybrid approaches using relatively simpler GRMs have been shown to be feasible in the continental to global scale mapping of flood hazards and risks in fine spatial resolution (e.g. Ward et al., 2013, Winsemius et al., 2013, and Dottori et al., 2016)." [page 12, lines 28-32]

*It would be good to see a sensitivity analysis of these model. How important are the sub-grid channel parameters in terms of the modelled output?*

Related comment: *Line 14, page 7: What are the confidence bounds in the values for Manning's coefficient? How do these confidence bounds impact on the model output? Is uncertainty in these values captured by other parameters?*

Thank you for asking about the sensitivity of the model to the sub-grid channel parameters. The sensitivities of the model to river bank height, channel width, and Manning's coefficient have been shown and discussed in Yamazaki et al., 2011. The sensitivity of the simulated river discharge and flooded area to the parameters are discussed in a full section in the said paper. To summarize the section, a deeper bank height, wider channel width, or smaller Manning's coefficient result in larger fluctuations and advanced peaks in the simulated discharge, and less flooded area. The topic is interesting but as it is already published, we decided not to include a sensitivity analysis in the current

paper to avoid duplication and to focus the discussion on the impacts of spatial resolution and representation of flow connectivity in the model.

To briefly discuss the topic in the paper, we have added the following in the appendix:

5 "The model is quite sensitive to the sub-grid channel parameters. A sensitivity analysis done by Yamazaki et al. (2011) showed that a deeper bank height, wider channel width, or smaller Manning's coefficient result in less flooded area and larger fluctuations and advanced peaks in simulated discharge." [page 16, lines 28-31]

As for the confidence bounds in the values of Manning's coefficient, we have found that the value used in this study is close to values published in other studies. Published values for river channels are: $0.03 - 0.035$ in Visutimeteegorn et al., 2007; $0.026 - 0.035$ in Keokhumcheng et al., 2010; $0.03 - 0.055$
10 in Tingsanchali and Karim, 2010; $0.03$ in Hunukumbura and Tachikawa, 2012; $0.03$ in Sayama et al., 2015; and $0.025 - 0.40$ in Wongsa, 2015. While the estimated value used in this study (0.024) is a little lower than the values published in other papers, it should be noted that some of these papers lumped the Manning's coefficient for the river channels and floodplains. Three of these published studies used a separate Manning's coefficient for plains and/or floodplains: $0.05 - 0.07$ in Visutimeteegorn et al.,
15 2007, $0.062$ in Keokhumcheng et al., 2010, and $0.35$ in Sayama et al., 2015. The estimated value for the Manning's coefficient in floodplains (0.10) used in this study falls within the published ranges.

Although we did not include an uncertainty or sensitivity analysis in this paper, for the purpose of applying the model in other river basins or regions, we believe that such analyses on the model parameters may be worth doing in the future.

*Lines 19 to 21, page 6: The authors might need to clarify this sentence a little. Does this mean that the CaMa-Flood values were adjusted to ensure that the catchment scale output matched the gridded runoff within the catchment? If so, how significant was this adjustment?*

Thank you for allowing us to clarify the sentence. As described in the text, we are using the same daily
25 runoff dataset for all experiments. The runoff dataset is available at a spatial resolution of 5-arcminute grids (approximately 10k). As we move to finer spatial resolutions, there will be an increase in the number of CaMa-Flood unit-catchments which will be found in an area covered by one runoff grid cell. To ensure that the mass of water between the gridded runoff inputs and CaMa-Flood is conserved, whenever the boundaries of a runoff grid cell do not match the boundaries of the CaMa-Flood unit-
30 catchments, area-weighted averaging is used to determine the amount of runoff that will have to be routed in each unit-catchment. The adjustment is made at the beginning of the simulation process to make sure that water is not created nor destroyed before being routed within CaMa-Flood. No adjustments are made to the outputs of the model. We hope our modified sentence is clearer:

"To conserve the mass of runoff inputs, CaMa-Flood uses area-weighted averaging to distribute the coarse,
35 gridded runoff among the unit-catchments in CaMa-Flood." [page 7, line 4 – 5]

*There are a significant number of typographical and grammatical errors in the paper (see the comments from the first reviewer for a fairly comprehensive list of these).*

*Line 28, page 3: I would suggestion "CaMa-Flood is currently the only GRM . . ."*

40 Thank you for the suggestion. We have made the necessary changes in the revised manuscript.

*Line 7, page 4: I think "can be answered" is a bit strong here (see comment above). Actually, all that has been done is investigate these questions from the point of view of a case study.*

Thank you for pointing this out. We have toned down the statement to "With this verification exercise, we attempt to answer two fundamental questions with regards to large- to global-scale simulation of floods…" [page 4, lines 11-12]

*Line 14, page 5: I think "Yamazaki et al., 2014b" is the wrong citation here. This makes it seem that the "local inertial equation" was first presented in this paper. A couple of lines later, the authors refer to a 2010 paper that talks about it. Options are to remove the citation (already given elsewhere in the paper), or give a more appropriate citation to the "local inertial equation".*

Thank you for pointing this out. We have changed the wording to "The latest and most efficient version of CaMa-Flood (Yamazaki et al., 2014b) which uses the "local inertial equation" (Bates et al., 2010)…" [page 5, lines 22-23]

We moved the citation of Yamazaki et al., 2014b next to "CaMa-Flood" and cited the first paper that presented the "local inertial equation." We think that it is necessary to cite "Yamazaki et al., 2014b" here to clearly identify which version of CaMa-Flood we are using.

*Line 16, page 5: either "represent the backwater effect" or "represents backwater effects" Line 27, page 5: either "the" or "a" before "bifurcation scheme"*

Thank you for the suggestion. We have made the necessary changes.

*Line 2, page 8: Are the objective functions evaluated at a site-by-site level, or are the aggregated across a number of sites. Judging by the rest of the text, it is done at a site level. If so, then there is a simple 1-to-1 relationship between NSE and RMSE, so using both would not give any additional information.*

Thank you for this comment. The referee is correct, the objective functions are evaluated at a site-by-site level. It is also true that there is a simple relationship between NSE and RMSE, and we see the point being made by the referee – that it may be redundant to show both metrics. Using NSE alone may be sufficient to prove our point – that the predictive capability of the model slightly increase with finer spatial resolution in the *MDC* scheme and significantly decrease with finer resolution in the *SDC* scheme. However, in some of the sites (gauging stations) that we have analysed, the differences in NSE coefficients between simulation results from 10km to 1km resolutions are usually smaller than those in RMSE. For example, in Figure 5, the upward trend in the NSE of discharge in C13_MDC (about 5% change) is less evident than the downward trend in the RMSE of discharge in the site (about 10% change). Hence, we find it useful to show both metrics in the paper.

*Line 3, page 8: Here the authors use "difference in discharge peak timing". Elsewhere (e.g. line 23, page 8; line 33, page 9) they use "delay". The authors should be consistent. In my view "difference" would be the better term to use as it is the difference in the delay in the peak that is being measured. Line 33, page 9: change "time to peaking" to "time to peak"*

Thank you for pointing this out. We have made the necessary changes and used "difference" consistently in the text.

*Line 4 to 7, page 12: There is an apparent contradiction in these two sentences. The first says that the*
5 *calibration of the sub-grid channel parameters is tedious – implying that this was a time consuming process. Later the authors state that the calibrated parameter values were found to be consistent and transferable across spatial resolutions – implying that calibration wasn't that difficult.*

Thank you for pointing out the confusing parts of the paper. Similar with most models that have multiple parameters, calibrating CaMa-Flood is time consuming. The amount of time required for
10 calibration significantly increases when we move from coarse to fine resolutions. In the test basin, we have found that the optimal parameters calibrated at a coarse spatial resolution (10k), do not significantly change when we move to finer spatial resolutions (as shown in the appendix and briefly mentioned in section 4). Hence, we can avoid the extra costs of re-calibrating the model at finer spatial resolutions (as the parameters were found to be transferable across spatial resolutions). Nonetheless,
15 time and effort still have to be spent to calibrate the model at some point.

Hopefully, the modified statements are clearer:

"One of the caveats of this study is the tedious calibration of the sub-grid channel parameters when applied to regional basins. At the global scale, this can potentially be addressed by the global width database available for large rivers (GWDLR developed by Yamazaki et al., 2014a). A database of channel depths of large rivers,
20 however, do not exist; hence, the parameters characterizing the channel depths in the model may still have to be calibrated. To ease the difficulty of calibration, the development of an automatic calibration tool or a simpler or more efficient parameterization of sub-grid channels (e.g. Neal et al., 2015) may be helpful.
In the test basin used in this study, it was found that the parameters calibrated at a coarse spatial resolution are transferable across finer spatial resolutions. This significantly reduces the time required to re-calibrate the model
25 at finer spatial resolutions. Once the initial difficulty of calibrating the necessary parameters at a coarse spatial resolution is hurdled, CaMa-Flood with *MDC* scheme can be used for more realistic, consistent, and robust simulation of large-scale floods across varying spatial resolutions. It should be noted, however, that the *MDC* scheme of CaMa-Flood had only been validated in three test basins – Mekong delta (Yamazaki et al., 2014b), Ganges-Brahmaputra-Meghna delta (Ikeuchi et al., 2015) and Chao Phraya River Basin in Thailand (this study).
30 It should also be noted that the transferability of calibrated parameters from a coarse spatial resolution to finer spatial resolutions have to be validated in other river basins. More extensive tests in large river basins and at global scale have to be conducted to further validate the model." [page 13, lines 2 – 16]

*Line 2, page 13: change "it significantly declined" to either "this significantly declined" or "it declined*
35 *significantly"*

Thank you for the suggestion. We have made the necessary changes in the revised manuscript.

References:

[revised manuscript text omitted]

---

## Author Response (AR2)

**Response to the Editor**

Dear editor, thank you very much for taking the time to carefully review our revised manuscript. We have further revised the manuscript to address the comments below:

I have assessed the manuscript as well as the Authors' replies to the Reviewers' comments/suggestions. From a personal standpoint, it is not clear how the Authors have quantified the uncertainty associated their estimated model parameters. Without a clear analysis of this aspect, results could be of indeterminate quality. Before making my final decision, I would like to Authors to provide a clear answer to this point. I am confident this would entail solely a set of minor revisions.

The suitability of the estimated model parameters for the river basin and for the numerical experiment in the study have been evaluated in two ways. Note that the pages and lines indicated in this document correspond to those in the revised manuscript without the changes tracked.

1) Suitability for use in the river basin

The model was calibrated and the methodology used for calibrating and validating the model is briefly discussed in Section 4 Calibration and Validation. We have improved this section by including the statistical results of the calibration (see page 7 line 27 to page 8 line 11):

"Based on Mateo et al. (2014), the Manning's coefficient was fixed at 0.024 in the river channel and 0.10 in the floodplain for the entire basin. These values of Manning's coefficient are comparable with those used in other studies in the Chao Phraya River Basin (Visutimeteegorn et al., 2007; Keokhumcheng et al., 2012; Sayama et al., 2015) and those obtained by USGS from lab and field data (Aldridge and Garrett, 1973). By using the calibrated parameters, CaMa-Flood can adequately simulate the discharge and flood inundation in the basin (Fig. 3) (Mateo et al., 2014).

The model calibrated at 10 km spatial resolution was found to have a good fit with observations. The discharge estimates from the model were in good agreement with the observed river discharge at the station used for calibration (C2 Station in Fig. 1), with the daily Nash-Sutcliffe efficiency coefficient (NSE) in year 2011 with SDC and MDC of 0.73 and 0.80, respectively. The Pearson correlation coefficients between the observed and model estimated discharge are very high (both above 0.90) and biases are low. There is also a very good agreement between the model estimated flood inundation extents and the satellite derived water maps for all available satellite images. Validation of the model for different years and other gauging stations in the Chao Phraya River Basin are also shown to be reasonable by Mateo et al. (2014). The results of the calibration confirm

that the parameterisation is reasonably robust and suitable for large scale application in the Chao Phraya River Basin. This is to be expected as even without calibration of the parameters, the use of CaMa-Flood with nine GHMs (which include the H08 model) results in better agreement with monthly to daily observations in 1701 globally distributed river discharge stations from the Global Runoff Data Center as compared with the native river routing schemes of the GHMs (Zhao et al., 2017)."

Further discussion can be found in the appendix and in a published paper (Mateo et al. 2014).

2) Suitability for use in finer spatial resolutions

We have verified that the parameters calibrated at 10km spatial resolution are also suitable for use in finer spatial resolutions. We have stressed this point at the end of Section 4 (see page 8 lines 12 to 16):

"The calibrated parameters were perturbed and applied at a finer spatial resolution to examine if recalibration or tuning of the parameters is needed to apply the model at finer spatial resolutions. It was found that the 'optimum' parameters do not significantly change with changes in spatial resolution (see Appendix). Hence, the parameters of the model which were calibrated at coarser scales can be applied at finer scales without recalibration to avoid huge computation overheads. The stability of the calibration across scales indicates that the model is robust."

We have improved the description as to how we verified the robustness of the calibrated parameters across spatial scales in the Appendix and have added the following lines (see page 17 lines 21 to 25):

"Table A1 shows that alternative parameter sets which give comparable results with the calibrated parameters exist (e.g. B7 or W3 as shown in Table A1). However, it should be noted that those parameter sets produce river widths and river bank heights that are within 10% difference in size as compared with those produced using the calibrated parameter set. Hence, the results confirm that the 'optimum' parameter set do not significantly change with spatial scale."

Note that we have improved Table A1 by providing the reasons for screening out parameter sets which are not suitable for use in the river basin. We have also updated the table to correct a few errors that have been overlooked in the previous versions of the paper.

Most importantly, we would like to clarify that the numerical experiments in this study have been repeated by using other parameter sets. For example, we have used (1) the global width database for large rives (GWDLR, Yamazaki et al. 2014a) to replace the calibrated river widths and (2) the default parameter sets of CaMa-Flood which are used for global studies. The results for those experiments are similar with those presented in this study (using calibrated parameters). We believe that this indicates

that our findings are robust and are independent of the parameters used in the model. We have included a paragraph to stress this point in Section 5 Results (see page 8 lines 26 to 32):

"While the results shown in the following subsections are obtained by running the model with calibrated parameters, numerical experiments were also conducted by running the model using alternative parameter values (e.g. parameter values used in global simulations, using the river widths from the global width database for large rivers, GWDLR by Yamazaki et al. 2014a). The findings remain the same (and thus are not shown in this paper for brevity) albeit there are differences in the magnitude of changes in model efficiency with changes in spatial resolution. As expected, the use of non-calibrated parameters resulted in larger changes in model efficiency. Hence, the findings of this study are robust and are independent of the parameters used in the model."

The revised manuscript with changes from the previous version tracked is attached at the end of this response.

References:

[revised manuscript text omitted]